# FedExP: Speeding Up Federated Averaging via Extrapolation

**Divyansh Jhunjhunwala[1], Shiqiang Wang[2], Gauri Joshi[1]**
[1]Carnegie Mellon University, [2]IBM Research
{djhunjhu, gaurij}@andrew.cmu.edu, wangshiq@us.ibm.com

## Abstract

Federated Averaging (`FedAvg`) remains the most popular algorithm for Federated Learning (FL) optimization due to its simple implementation, stateless nature, and privacy guarantees combined with secure aggregation. Recent work has sought to generalize the vanilla averaging in `FedAvg` to a generalized gradient descent step by treating client updates as pseudo-gradients and using a server step size. While the use of a server step size has been shown to provide performance improvement theoretically, the practical benefit of the server step size has not been seen in most existing works. In this work, we present `FedExP`, a method to adaptively determine the server step size in FL based on dynamically varying pseudo-gradients throughout the FL process. We begin by considering the overparameterized convex regime, where we reveal an interesting similarity between `FedAvg` and the Projection Onto Convex Sets (`POCS`) algorithm. We then show how `FedExP` can be motivated as a novel extension to the *extrapolation mechanism* that is used to speed up POCS. Our theoretical analysis later also discusses the implications of `FedExP` in underparameterized and non-convex settings. Experimental results show that `FedExP` consistently converges faster than `FedAvg` and competing baselines on a range of realistic FL datasets.

## 1 Introduction

Federated Learning (FL) has emerged as a key distributed learning paradigm in which a central server orchestrates the training of a machine learning model across a network of devices. FL is based on the fundamental premise that data never leaves a clients device, as clients only communicate model updates with the server. Federated Averaging or `FedAvg`, first introduced by McMahan et al. (2017), remains the most popular algorithm in this setting due to the simplicity of its implementation, stateless nature (i.e., clients do not maintain local parameters during training) and the ability to incorporate privacy-preserving protocols such as secure aggregation (Bonawitz et al., 2016; Kadhe et al., 2020).

**Slowdown Due to Heterogeneity.** One of the most persistent problems in `FedAvg` is the slowdown in model convergence due to data heterogeneity across clients. Clients usually perform multiple steps of gradient descent on their heterogeneous objectives before communicating with the server in `FedAvg`, which leads to what is colloquially known as *client drift error* (Karimireddy et al., 2019). The effect of heterogeneity is further exacerbated by the constraint that only a fraction of the total number of clients may be available for training in every round (Kairouz et al., 2021). Various techniques have been proposed to combat this slowdown, among the most popular being variance reduction techniques such as Karimireddy et al. (2019); Mishchenko et al. (2022); Mitra et al. (2021), but they either lead to clients becoming stateful, add extra computation or communication requirements or have privacy limitations.

**Server Step Size.** Recent work has sought to deal with this slowdown by using two separate step sizes in `FedAvg` – a *client step size* used by the clients to minimize their local objectives and a *server step size* used by the server to update the global model by treating client updates as pseudo-gradients (Karimireddy et al., 2019; Reddi et al., 2021). To achieve the fastest convergence rate, these works propose keeping the client step size as $\mathcal{O}(1/\tau\sqrt{T})$ and the server step size as $\mathcal{O}(\sqrt{\tau M})$, where $T$ is the number of communication rounds, $\tau$ is the number of local steps and $M$ is the number of clients. Using a small client step size mitigates client drift, and a large server

step size prevents global slowdown. While this idea may be asymptotically optimal, it is not always effective in practical non-asymptotic and communication-limited settings (Charles & Konečný, 2020). In practice, a small client step size severely slows down convergence in the initial rounds and cannot be fully compensated for by a large server step size (see Figure 1). Also, if local objectives differ significantly, then it may be beneficial to use smaller values of the server step size (Malinovsky et al., 2022).

Therefore, we seek to answer the following question: *For a moderate client step size, can we adapt the server step size according to the local progress made by the clients and the heterogeneity of their objectives?* In general, it is challenging to answer this question because it is difficult to obtain knowledge of the heterogeneity between the local objectives and appropriately use it to adapt the server step size.

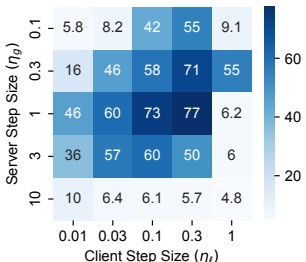

Figure 1: Test accuracy (%) achieved by different server and client step sizes on EMNIST dataset (Cohen et al., 2017) after 50 rounds (details of experimental setup are in Section 6 and Appendix D).

**Our Contributions.** In this paper, we take a novel approach to address the question posed above. We begin by considering the case where the models are *overparameterized*, i.e., the number of model parameters is larger than the total number of data points across all clients. This is often true for modern deep neural network models (Zhang et al., 2017; Jacot et al., 2018) and the small datasets collected by edge clients in the FL setting. In this overparameterized regime, the global minimizer becomes a *common minimizer* for all local objectives, even though they may be arbitrarily heterogeneous. Using this fact, we obtain a *novel connection* between `FedAvg` and the Projection Onto Convex Sets (`POCS`) algorithm, which is used to find a point in the intersection of some convex sets.

Based on this connection, we find an interesting analogy between the server step size and the *extrapolation parameter* that is used to speed up `POCS` (Pierra, 1984). We propose new extensions to the extrapolated `POCS` algorithm to support inexact and noisy projections as in `FedAvg`. In particular, we derive a *time-varying* bound on the progress made by clients towards the global minimum and show how this bound can be used to adaptively estimate a good server step size at each round. The result is our proposed algorithm `FedExP`, which is a method to adaptively determine the server step size in each round of FL based on the pseudo-gradients in that round.

Although motivated by the overparameterized regime, our proposed `FedExP` algorithm performs well (both theoretically and empirically) in the general case, where the model can be either overparameterized or underparameterized. For this general case, we derive the convergence upper bounds for both convex and non-convex objectives. Some highlights of our work are as follows.

- We reveal a novel connection between `FedAvg` and the `POCS` algorithm for finding a point in the intersection of convex sets.

- The proposed `FedExP` algorithm is simple to implement with virtually no additional communication, computation, or storage required at clients or the server. It is well suited for both cross-device and cross-silo FL, and is compatible with partial client participation.

- Experimental results show that `FedExP` converges 1.4–2× faster than `FedAvg` and most competing baselines on standard FL tasks.

**Related Work.** Popular algorithms for adaptively tuning the step size when training neural networks include `Adagrad` (Duchi et al., 2011) and its variants `RMSProp` (Tieleman et al., 2012) and `Adadelta` (Zeiler, 2012). These algorithms consider the notion of *coordinate-wise adaptivity* and adapt the step size separately for each dimension of the parameter vector based on the magnitude of the accumulated gradients. While these algorithms can be extended to the federated setting using the concept of pseudo-gradients as done by Reddi et al. (2021), these extensions are agnostic to inherent data heterogeneity across clients, which is central to FL. On the contrary, `FedExP` is explicitly designed for FL settings and uses a *client-centric* notion of adaptivity that utilizes the heterogeneity of client updates in each round. The work closest to us is Johnson et al. (2020), which proposes a method to adapt the step size for large-batch training by estimating the *gradient diversity* (Yin et al., 2018) of a minibatch. This result has been improved in a recent work by Horváth et al. (2022). However, both Johnson et al. (2020); Horváth et al. (2022) focus on the centralized setting. In

`FedExP`, we use a similar concept, but within a *federated* environment which comes with a stronger theoretical motivation, since client data are inherently diverse in this case. We defer a more detailed discussion of other adaptive step size methods and related work to Appendix A.

## 2 PROBLEM FORMULATION AND PRELIMINARIES

As in most standard federated learning frameworks, we consider the problem of optimizing the model parameters $\mathbf{w} \in \mathbb{R}^d$ to minimize the global objective function $F(\mathbf{w})$ defined as follows:

$$\min_{\mathbf{w} \in \mathbb{R}^d} F(\mathbf{w}) := \frac{1}{M} \sum_{i=1}^{M} F_i(\mathbf{w}), \tag{1}$$

where $F_i(\mathbf{w}) := \frac{1}{|\mathcal{D}_i|} \sum_{\delta_i \in \mathcal{D}_i} \ell(\mathbf{w}, \delta_i)$ is the empirical risk objective computed on the local data set $\mathcal{D}_i$ at the the $i$-th client. Here, $\ell(\cdot, \cdot)$ is a loss function and $\delta_i$ represents a data sample from the empirical local data distribution $\mathcal{D}_i$. The total number of clients in the FL system is denoted by $M$. Without loss of generality, we assume that all the $M$ client objectives are given equal weight in the global objective function defined in (1). Our algorithm and analysis can be directly extended to the case where client objectives are unequally weighted, e.g., proportional to local dataset sizes $|\mathcal{D}_i|$.

**FedAvg.** We focus on solving (1) using `FedAvg` (McMahan et al., 2017; Kairouz et al., 2021). At round $t$ of `FedAvg`, the server sends the current global model $\mathbf{w}^{(t)}$ to all clients. Upon receiving the global model, clients perform $\tau$ steps of local stochastic gradient descent (SGD) to compute their updates $\{\Delta_i^{(t)}\}_{i=1}^{M}$ for round $t$ as follows.

$$\text{Perform Local SGD:} \quad \mathbf{w}_i^{(t,k+1)} = \mathbf{w}_i^{(t,k)} - \eta_l \nabla F_i(\mathbf{w}_i^{(t,k)}, \xi^{(t,k)}) \quad \forall k \in \{0, 1, \dots, \tau - 1\} \tag{2}$$

$$\text{Compute Local Difference:} \quad \Delta_i^{(t)} = \mathbf{w}^{(t)} - \mathbf{w}_i^{(t,\tau)} \tag{3}$$

where $\mathbf{w}_i^{(t,0)} = \mathbf{w}^{(t)}$ for all $i \in [M]$, $\eta_l$ is the client step size and $\nabla F_i(\mathbf{w}_i^{(t,k)}, \xi^{(t,k)})$ represents a stochastic gradient computed on the minibatch $\xi_i^{(t,k)}$ sampled randomly from $\mathcal{D}_i$.

**Server Optimization in `FedAvg`.** In vanilla `FedAvg` (McMahan et al., 2017), the global model would simply be updated as the average of the client local models, that is, $\mathbf{w}^{(t+1)} = \frac{1}{M} \sum_{i=1}^{M} \mathbf{w}_i^{(t,\tau)}$. To improve over this, recent work (Reddi et al., 2021; Hsu et al., 2019) has focused on optimizing the server aggregation process by treating the client updates $\Delta_i^{(t)}$ as "pseudo-gradients" and multiplying by a server step size when aggregating them as follows.

$$\text{Generalized \texttt{FedAvg} Global Update:} \quad \mathbf{w}^{(t+1)} = \mathbf{w}^{(t)} - \eta_g \bar{\Delta}^{(t)} \tag{4}$$

where $\bar{\Delta}^{(t)} = \frac{1}{M} \sum_{i=1}^{M} \Delta_i^{(t)}$ is the aggregated client update in round $t$ and $\eta_g$ acts as *server step size*. Note that setting $\eta_g = 1$ recovers the vanilla `FedAvg` update.

While the importance of the server step size has been theoretically well established in these works, we find that its practical relevance has not been explored. In this work, we take a step towards bridging this gap between theory and practice by *adaptively* tuning the value of $\eta_g$ that we use in every round.

## 3 PROPOSED ALGORITHM: FEDEXP

Before discussing our proposed algorithm, we first highlight a useful and novel connection between `FedAvg` and the `POCS` algorithm used to find a point in the intersection of some convex sets.

### 3.1 MOTIVATION FOR EXTRAPOLATION

**Connection Between `FedAvg` and `POCS` in the Overparameterized Convex Regime.** Consider the case where the local objectives of the clients $\{F_i(\mathbf{w})\}_{i=1}^{M}$ are convex. In this case, we know that the set of minimizers of $F_i(\mathbf{w})$ given by $\mathcal{S}_i^* = \{\mathbf{w} : \mathbf{w} \in \arg\min F_i(\mathbf{w})\}$ is also a convex set for all $i \in [M]$. Now let us assume that we are in the *overparameterized regime* where $d$ is sufficiently larger than the total number of data points across clients. In this regime, the model can

fit all the training data at clients simultaneously and hence be a minimizer for all local objectives. Thus we assume that the global minimum satisfies $\mathbf{w}^* \in \mathcal{S}_i^*, \forall i \in [M]$. Our original problem in (1) can then be reformulated as trying to *find a point in the intersection of convex sets* $\{\mathcal{S}_i^*\}_{i=1}^M$ since $\mathbf{w}^* \in \mathcal{S}_i^*, \forall i \in [M]$. One of the most popular algorithms to do so is the *Projection Onto Convex Sets* (POCS) algorithm (Gurin et al., 1967). In POCS, at every iteration the current model is updated as follows[1].

Generalized POCS update:
$$\mathbf{w}_{\text{POCS}}^{(t+1)} = \mathbf{w}_{\text{POCS}}^{(t)} - \lambda \left( \frac{1}{M} \sum_{i=1}^M P_i\big(\mathbf{w}_{\text{POCS}}^{(t)}\big) - \mathbf{w}_{\text{POCS}}^{(t)} \right) \quad (5)$$

where $P_i\big(\mathbf{w}_{\text{POCS}}^{(t)}\big)$ is a projection of $\mathbf{w}_{\text{POCS}}^{(t)}$ on the set $\mathcal{S}_i^*$ and $\lambda$ is known as the relaxation coefficient (Combettes, 1997).

**Extrapolation in POCS.** Combettes (1997) notes that POCS has primarily been used with $\lambda = 1$, with studies failing to demonstrate a systematic benefit of $\lambda < 1$ or $\lambda > 1$ (Mandel, 1984). This prompts Combettes (1997) to study an *adaptive* method of setting $\lambda$, first introduced by Pierra (1984) as follows:

$$\lambda^{(t)} = \frac{\sum_{i=1}^M \left\| P_i(\mathbf{w}^{(t)}) - \mathbf{w}^{(t)} \right\|^2}{M \left\| \frac{1}{M} \sum_{i=1}^M P_i(\mathbf{w}^{(t)}) - \mathbf{w}^{(t)} \right\|^2} .$$

Pierra (1984) refer to the POCS algorithm with this adaptive $\lambda^{(t)}$ as Extrapolated Parallel Projection Method (EPPM). This is referred to as extrapolation since we always have $\lambda^{(t)} \geq 1$ by Jensen's inequality. The intuition behind EPPM lies in showing that the update with the proposed $\lambda^{(t)}$ always satisfies $\left\| \mathbf{w}_{\text{POCS}}^{(t+1)} - \mathbf{w}^* \right\|^2 < \left\| \mathbf{w}_{\text{POCS}}^{(t)} - \mathbf{w}^* \right\|^2$, thereby achieving asymptotic convergence. Experimental results in Pierra (1984) and Combettes (1997) show that EPPM can give an order-wise speedup over POCS, motivating us to study this algorithm in the FL context.

## 3.2 Incorporating Extrapolation in FL

Note that to implement POCS we do not need to explicitly know the sets $\{\mathcal{S}_i^*\}_{i=1}^M$; we only need to know how to compute a *projection* on these sets. From this point of view, we see that FedAvg proceeds similarly to POCS. In each round, clients receive $\mathbf{w}^{(t)}$ from the server and run multiple SGD steps to compute an "approximate projection" $\mathbf{w}_i^{(t,\tau)}$ of $\mathbf{w}^{(t)}$ on their solution sets $\mathcal{S}_i^*$. These approximate projections are then aggregated at the server to update the global model. In this case, the relaxation coefficient $\lambda$ plays exactly the same role as the server step size $\eta_g$ in FedAvg.

Inspired by this observation and the idea of extrapolation in POCS, we seek to understand if a similar idea can be applied to tune the server step size $\eta_g$ in FedAvg. Note that the EPPM algorithm makes use of exact projections to prove convergence which is not available to us in FL settings. This is further complicated by the fact that the client updates are noisy due to the stochasticity in sampling minibatches. We find that in order to use an EPPM-like step size the use of exact projections can be relaxed to the following condition, which bounds the distance of the local models from the global minimum as follows.

Approximate projection condition in FL:
$$\frac{1}{M} \sum_{i=1}^M \left\| \mathbf{w}_i^{(t,\tau)} - \mathbf{w}^* \right\|^2 \leq \left\| \mathbf{w}^{(t)} - \mathbf{w}^* \right\|^2 \quad (6)$$

where $\mathbf{w}^{(t)}$ and $\{\mathbf{w}_i^{(t,\tau)}\}_{i=1}^M$ are the global and local client models, respectively, at round $t$ and $\mathbf{w}^*$ is a global minimum. Intuitively, this condition suggests that after the local updates, the local models are closer to the optimum $\mathbf{w}^*$ on average as compared to model $\mathbf{w}^{(t)}$ at the beginning of that round. We first show that this condition (6) holds in the overparameterized convex regime under some conditions. The full proofs for lemmas and theorems in this paper are included in Appendix C.

**Lemma 1.** *Let $F_i(\mathbf{w})$ be convex and $L$-smooth for all $i \in [M]$ and let $\mathbf{w}^*$ be a common minimizer of all $F_i(\mathbf{w})$. Assuming clients run full-batch gradient descent to minimize their local objectives with $\eta_l \leq 1/L$, then (6) holds for all $t$ and $\tau \geq 1$.*

In the case with stochastic gradient noise or when the model is underparameterized, although (6) may not hold in general, we expect it to be satisfied at least during the initial phase of training when $\left\| \mathbf{w}^{(t)} - \mathbf{w}^* \right\|^2$ is large and clients make common progress towards a minimum.

---

[1] We refer here to a parallel implementation of POCS. This is also known as Parallel Projection Method (PPM) and Simultaneous Iterative Reconstruction Technique (SIRT) in some literature (Combettes, 1997).

---

**Algorithm 1** Proposed Algorithm: `FedExP`

---

1: **Input:** $\mathbf{w}^{(0)}$, number of rounds $T$, local iteration steps $\tau$, parameters $\eta_l, \epsilon$
2: **For** $t = 0, \ldots, T - 1$ **communication rounds do**:
3:     **Global server does:**
4:     Send $\mathbf{w}^{(t)}$ to all clients
5:     **Clients** $i \in [M]$ **in parallel do:**
6:         Set $\mathbf{w}_i^{(t,0)} \leftarrow \mathbf{w}^{(t,0)}$
7:         **For** $k = 0, \ldots, \tau - 1$ **local iterations do:**
8:             Update $\mathbf{w}_i^{(t,k+1)} \leftarrow \mathbf{w}_i^{(t,k)} - \eta_l \nabla F_i(\mathbf{w}_i^{(t,k)}, \xi_i^{(t,k)})$
9:         Send $\Delta_i^{(t)} \leftarrow \mathbf{w}^{(t)} - \mathbf{w}_i^{(t,\tau)}$ to the server
10:    **Global server does:**
11:    Compute $\bar{\Delta}^{(t)} \leftarrow \frac{1}{M} \sum_{i=1}^{M} \Delta_i^{(t)}$ and $\eta_g^{(t)} \leftarrow \max\left\{ 1, \sum_{i=1}^{M} \left\| \Delta_i^{(t)} \right\|^2 \Big/ 2M\left( \left\| \bar{\Delta}^{(t)} \right\|^2 + \epsilon \right) \right\}$
12:    Update global model with $\mathbf{w}^{(t+1)} \leftarrow \mathbf{w}^{(t)} - \eta_g^{(t)} \bar{\Delta}^{(t)}$

---

Given that (6) holds, we now consider the generalized `FedAvg` update with a server step size $\eta_g^{(t)}$ in round $t$. Our goal is to find the value of $\eta_g^{(t)}$ that *minimizes* the distance of $\mathbf{w}^{(t+1)}$ to $\mathbf{w}^*$:

$$\left\| \mathbf{w}^{(t+1)} - \mathbf{w}^* \right\|^2 = \left\| \mathbf{w}^{(t)} - \mathbf{w}^* \right\|^2 + (\eta_g^{(t)})^2 \left\| \bar{\Delta}^{(t)} \right\|^2 - 2\eta_g^{(t)} \left\langle \mathbf{w}^{(t)} - \mathbf{w}^*, \bar{\Delta}^{(t)} \right\rangle. \tag{7}$$

Setting the derivative of the RHS of (7) to zero we have,

$$(\eta_g^{(t)})_{\text{opt}} = \frac{\left\langle \mathbf{w}^{(t)} - \mathbf{w}^*, \bar{\Delta}^{(t)} \right\rangle}{\left\| \bar{\Delta}^{(t)} \right\|^2} = \frac{\sum_{i=1}^{M} \left\langle \mathbf{w}^{(t)} - \mathbf{w}^*, \Delta_i^{(t)} \right\rangle}{M \left\| \bar{\Delta}^{(t)} \right\|^2} \geq \frac{\sum_{i=1}^{M} \left\| \Delta_i^{(t)} \right\|^2}{2M \left\| \bar{\Delta}^{(t)} \right\|^2}, \tag{8}$$

where the last inequality follows from $\langle \mathbf{a}, \mathbf{b} \rangle = \frac{1}{2}[\|\mathbf{a}\|^2 + \|\mathbf{b}\|^2 - \|\mathbf{a} - \mathbf{b}\|^2]$, definition of $\Delta_i^{(t)}$ in (3) and (6). Note that depending on the values of $\{\Delta_i^{(t)}\}_{i=0}^{M}$, we may have $(\eta_g^{(t)})_{\text{opt}} \gg 1$. Thus, we see that (6) acts as a suitable replacement for projection to justify the use of extrapolation in FL settings.

## 3.3 PROPOSED ALGORITHM

Motivated by our findings above, we propose the following server step size for the generalized `FedAvg` update at each round:

$$(\eta_g^{(t)})_{\text{FedExP}} = \max\left\{ 1, \frac{\sum_{i=1}^{M} \left\| \Delta_i^{(t)} \right\|^2}{2M(\left\| \bar{\Delta}^{(t)} \right\|^2 + \epsilon)} \right\}. \tag{9}$$

We term our algorithm *Federated Extrapolated Averaging* or `FedExP`, in reference to the original `EPPM` algorithm which inspired this work. Note that our proposed step size satisfies the property that $\left| (\eta_g^{(t)})_{\text{opt}} - (\eta_g^{(t)})_{\text{FedExP}} \right| \leq \left| (\eta_g^{(t)})_{\text{opt}} - 1 \right|$ when (6) holds, which can be seen by comparing (8) and (9). Since (7) depends quadratically on $\eta_g^{(t)}$, we can show that in this case $\left\| \mathbf{w}^{(t+1)} - (\eta_g^{(t)})_{\text{FedExP}} \bar{\Delta}^{(t)} - \mathbf{w}^* \right\|^2 \leq \left\| \mathbf{w}^{(t+1)} - \mathbf{w}^* \right\|^2$, implying we are at least as close to the optimum as the `FedAvg` update. In the rest of the paper, we denote $(\eta_g^{(t)})_{\text{FedExP}}$ as $\eta_g^{(t)}$ when the context is clear.

**Importance of Adding Small Constant to Denominator.** In the case where (6) does not hold, using the lower bound established in (8) can cause the proposed step size to blow up. This is especially true towards the end of training where we can have $\left\| \bar{\Delta}^{(t)} \right\|^2 \approx 0$ but $\left\| \Delta_i^{(t)} \right\|^2 \neq 0$. Thus we propose to add a small positive constant $\epsilon$ to the denominator in (9) to prevent this blow-up. For a large enough $\epsilon$ our algorithm reduces to `FedAvg` and therefore tuning $\epsilon$ can be a useful tool to interpolate between vanilla averaging and extrapolation. Similar techniques exist in adaptive algorithms such as `Adam` (Kingma & Ba, 2015) and `Adagrad` (Duchi et al., 2011) to improve stability.

**Compatibility with Partial Client Participation and Secure Aggregation.** Note that `FedExP` can be easily extended to support partial participation of clients by calculating $\eta_g^{(t)}$ using only the updates of participating clients, i.e., the averaging and division in (9) will be only over the clients that participate in the round. Furthermore, since the server only needs to estimate the average of pseudo-gradient norms, $\eta_g^{(t)}$ can be computed with secure aggregation, similar to computing $\bar{\Delta}^{(t)}$.

**Connection with Gradient Diversity.** We see that our lower bound on $(\eta_g^{(t)})_{\text{opt}}$ naturally depends on the similarity of the client updates with each other. In the case where $\tau = 1$ and clients run full-batch gradient descent, our lower bound (8) reduces to $\sum_{i=1}^{M} \left\| \nabla F_i(\mathbf{w}^{(t)}) \right\|^2 / 2M \left\| \nabla F(\mathbf{w}^{(t)}) \right\|^2$ which is used as a measure of data-heterogeneity in many FL works (Wang et al., 2020; Haddadpour & Mahdavi, 2019). Our lower bound suggests using larger step-sizes as this gradient diversity increases, which can be a useful tool to speed up training in heterogeneous settings. This is an orthogonal approach to existing optimization methods to tackle heterogeneity such as Karimireddy et al. (2020b); Li et al. (2020); Acar et al. (2021), which propose additional regularization terms or adding control variates to the local client objectives to limit the impact of heterogeneity.

## 4 CONVERGENCE ANALYSIS

Our analysis so far has focused on the overparameterized convex regime to motivate our algorithm. In this section we discuss the convergence of our algorithm in the presence of underparameterization and non-convexity. We would like to emphasize that (6) is *not needed* to show convergence of `FedExP`; it is only needed to motivate why `FedExP` might be beneficial. To show general convergence, we only require that $\eta_l$ be sufficiently small and the standard assumptions stated below.

**Challenge in incorporating stochastic noise and partial participation.** Our current analysis focuses on the case where clients are computing full-batch gradients in every step with full participation. This is primarily due to the difficulty in decoupling the effect of stochastic and sampling noise on $\eta_g^{(t)}$ and the pseudo-gradients $\{\Delta_i^{(t)}\}_{i=1}^{M}$. To be more specific, if we use $\xi^{(t)}$ to denote the randomness at round $t$, then $\mathbb{E}_{\xi^{(t)}} \left[ (\eta_g^{(t)}) \bar{\Delta}^{(t)} \right] \neq \mathbb{E}_{\xi^{(t)}} \left[ (\eta_g^{(t)}) \right] \mathbb{E}_{\xi^{(t)}} \left[ \bar{\Delta}^{(t)} \right]$ which significantly complicates the proof. This is purely a theoretical limitation. Empirically, our results in Section 6 show that `FedExP` performs well with both SGD and partial client participation.

**Assumption 1.** *(L-smoothness) Local objective $F_i(\mathbf{w})$ is differentiable and L-smooth for all $i \in [M]$, i.e., $\|\nabla F_i(\mathbf{w}) - \nabla F_i(\mathbf{w}')\| \leq L\|\mathbf{w} - \mathbf{w}'\|, \forall \mathbf{w}, \mathbf{w}' \in \mathbb{R}^d$.*

**Assumption 2.** *(Bounded data heterogenenity at optimum) The norm of the client gradients at the global optima $\mathbf{w}^*$ is bounded as follows: $\frac{1}{M} \sum_{i=1}^{M} \|\nabla F_i(\mathbf{w}^*)\|^2 \leq \sigma_*^2$.*

**Theorem 1.** *($F_i$ are convex) Under Assumptions 1,2 and assuming clients compute full-batch gradients with full participation and $\eta_l \leq \frac{1}{6\tau L}$, the iterates $\{\mathbf{w}^{(t)}\}$ generated by `FedExP` satisfy,*

$$F(\bar{\mathbf{w}}^{(T)}) - F^* \leq \underbrace{\mathcal{O}\left( \frac{\left\| \mathbf{w}^{(0)} - \mathbf{w}^* \right\|^2}{\eta_l \tau \sum_{t=0}^{T-1} \eta_g^{(t)}} \right)}_{T_1 := \text{initialization error}} + \underbrace{\mathcal{O}\left( \eta_l^2 \tau(\tau-1) L \sigma_*^2 \right)}_{T_2 := \text{client drift error}} + \underbrace{\mathcal{O}\left( \eta_l \tau \sigma_*^2 \right)}_{T_3 := \text{noise at optimum}}, \quad (10)$$

*where $\eta_g^{(t)}$ is the `FedExP` server step size at round $t$ and $\bar{\mathbf{w}}^{(T)} = \frac{\sum_{t=0}^{T-1} \eta_g^{(t)} \mathbf{w}^{(t)}}{\sum_{t=0}^{T-1} \eta_g^{(t)}}$.*

For the non-convex case, we need the data heterogeneity to be bounded everywhere as follows.

**Assumption 3.** *(Bounded global gradient variance) There exists a constant $\sigma_g^2 > 0$ such that the global gradient variance is bounded as follows. $\frac{1}{M} \sum_{i=1}^{M} \|\nabla F_i(\mathbf{w}) - \nabla F(\mathbf{w})\|^2 \leq \sigma_g^2, \forall \mathbf{w} \in \mathbb{R}^d$.*

**Theorem 2.** *($F_i$ are non-convex) Under Assumptions 1, 3 and assuming clients compute full-batch gradients with full participation and $\eta_l \leq \frac{1}{6\tau L}$, the iterates $\{\mathbf{w}^{(t)}\}$ generated by `FedExP` satisfy,*

$$\min_{t \in [T]} \left\| \nabla F(\mathbf{w}^{(t)}) \right\|^2 \leq \underbrace{\mathcal{O}\left( \frac{F(\mathbf{w}^{(0)}) - F^*}{\eta_l \tau \sum_{t=0}^{T-1} \eta_g^{(t)}} \right)}_{T_1 := \text{initialization error}} + \underbrace{\mathcal{O}\left( \eta_l^2 L^2 (\tau-1)\tau \sigma_g^2 \right)}_{T_2 := \text{client drift error}} + \underbrace{\mathcal{O}\left( \eta_l L \tau \sigma_g^2 \right)}_{T_3 := \text{global variance}}, \quad (11)$$

*where $\eta_g^{(t)}$ is the `FedExP` server step size at round $t$.*

**Discussion.** In the convex case, the error of `FedAvg` can be bounded by $\mathcal{O}\left( \|\mathbf{w}^{(0)} - \mathbf{w}^*\|^2 / \eta_l \tau T \right) + \mathcal{O}\left( \eta_l^2 \tau(\tau-1) L \sigma_*^2 \right)$ (Khaled et al., 2020) and in the non-convex case by $\mathcal{O}\left( (F(\mathbf{w}^0) - F^*)/\eta_l \tau T \right) + \mathcal{O}\left( \eta_l^2 L^2 \tau(\tau-1) \sigma_g^2 \right)$ (Wang et al., 2020). A careful inspection

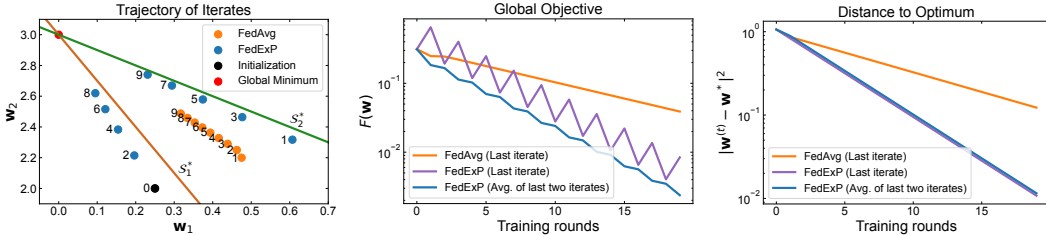

Figure 2: Training characteristics of `FedAvg` and `FedExP` for the 2-D toy problem in Section 5. The last iterate of `FedExP` has an oscillating behavior in $F(\mathbf{w})$ but monotonically decreases $\left\| \mathbf{w}^{(t)} - \mathbf{w}^* \right\|^2$; the average of the last two iterates lies in a lower loss region than the last iterate.

reveals that the impact of $T_1$ on convergence of `FedExP` is different from `FedAvg` (effect of $T_2$ is the same). We see that since $\sum_{t=0}^{T-1} \eta_g^{(t)} \geq T$, `FedExP` reduces $T_1$ faster than `FedAvg`. However this comes at the price of an increased error floor due to $T_3$. Thus, the larger step-sizes in `FedExP` help us reach the vicinity of an optimum faster, but can ultimately end up saturating at a higher error floor due to noise around the optimum. Note that the impact of the error floor can be controlled by setting the client step size $\eta_l$ appropriately. Moreover, in the overparameterized convex regime where $\sigma_*^2 = 0$, the effect of $T_2$ and $T_3$ vanishes and thus `FedExP` clearly outperforms `FedAvg`. This aligns well with our initial motivation of using extrapolation in the overparameterized regime.

## 5 FURTHER INSIGHTS INTO FEDEXP

In this section, we discuss some further insights into the training of `FedExP` and how we leverage these insights to improve the performance of `FedExP`.

**`FedExP` monotonically decreases $\left\| \mathbf{w}^{(t)} - \mathbf{w}^* \right\|^2$ but not necessarily $F(\mathbf{w}^{(t)}) - F(\mathbf{w}^*)$.** Recall that our original motivation for the `FedExP` step size was aimed at trying to minimize the *distance* to the optimum give by $\left\| \mathbf{w}^{(t+1)} - \mathbf{w}^* \right\|^2$, when (6) holds. Doing so satisfies $\left\| \mathbf{w}^{(t+1)} - \mathbf{w}^* \right\|^2 \leq \left\| \mathbf{w}^{(t)} - \mathbf{w}^* \right\|^2$ but does not necessarily satisfy $F(\mathbf{w}^{(t+1)}) \leq F(\mathbf{w}^{(t)})$.

To better illustrate this phenomenon, we consider the following toy example in $\mathbb{R}^2$. We consider a setup with two clients, where the objective at each client is given as follows:

$$F_1(\mathbf{w}) = (3w_1 + w_2 - 3)^2; \quad F_2(\mathbf{w}) = (w_1 + w_2 - 3)^2. \tag{12}$$

We denote the set of minimizers of $F_1(\mathbf{w})$ and $F_2(\mathbf{w})$ by $\mathcal{S}_1^* = \{\mathbf{w} : 3w_1 + w_2 = 3\}$ and $\mathcal{S}_2^* = \{\mathbf{w} : w_1 + w_2 = 3\}$ respectively. Note that $\mathcal{S}_1^*$ and $\mathcal{S}_2^*$ intersect at the point $\mathbf{w}^* = [0, 3]$, making it a global minimum. To minimize their local objectives, we assume clients run gradient descent with $\tau \to \infty$ in every round[2]. Figure 2 shows the trajectory of the iterates generated by `FedExP` and `FedAvg`. We see that while $\left\| \mathbf{w}^{(t)} - \mathbf{w}^* \right\|^2$ decreases monotonically for `FedExP`, $F(\mathbf{w}^{(t)})$ does not do so and in fact has an oscillating nature as we discuss below.

**Understanding oscillations in $F(\mathbf{w}^{(t)})$.** We see that the oscillations in $F(\mathbf{w}^{(t)})$ are caused by `FedExP` iterates trying to minimize their distance from the solution sets $\mathcal{S}_1^*$ and $\mathcal{S}_2^*$ simultaneously. The initialization point $\mathbf{w}^{(0)}$ is closer to $\mathcal{S}_1^*$ than $\mathcal{S}_2^*$, which causes the `FedExP` iterate at round 1 to move towards $\mathcal{S}_2^*$, then back towards $\mathcal{S}_1^*$ and so on. To understand why this happens, consider the case where $\Delta_1^{(t)} = 0, \Delta_2^{(t)} \neq 0$. In this case, we have $\eta_g^{(t)} = 2$ and therefore $\mathbf{w}^{(t+1)} = \mathbf{w}^{(t)} - 2\bar{\Delta}^{(t)} = \mathbf{w}_2^{(t,\tau)}$, which indicates that `FedExP` is now trying to minimize $\left\| \Delta_2^{(t+1)} \right\|^2$. This gives us the intuition that the `FedExP` update in round $t$ is trying to minimize the objectives of the clients that have $\left\| \Delta_i^{(t)} \right\|^2 \gg 0$. While this leads to a temporary increase in global loss $F(\mathbf{w}^{(t)})$ in

---

[2]The local models will be an exact projection of the global model on the solution sets $\{\mathcal{S}_i^*\}_{i=1}^2$. In this case, the lower bound in (8) can be improved by a factor of 2 and therefore we use $\eta_g^{(t)} = (\|\Delta_1\|^2 + \|\Delta_2\|^2)/2\|\bar{\Delta}^{(t)}\|^2$ for this experiment (see Appendix C.4 and Appendix C.4.1 for proof).

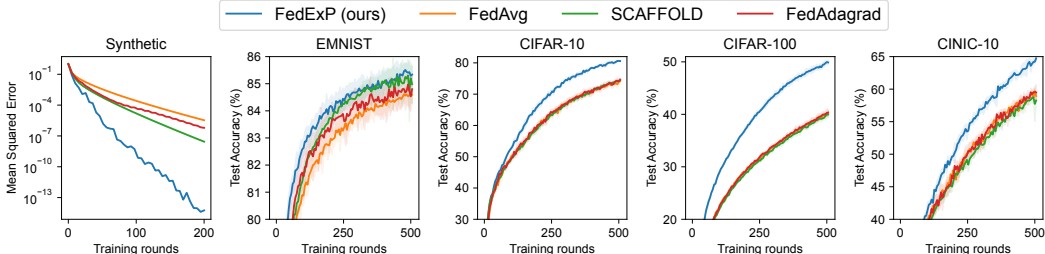

Figure 3: Experimental results on a synthetic linear regression experiments and a range of realistic FL tasks. FedExP consistently gives faster convergence compared to baselines while adding no extra computation, communication or storage at clients or server.

some rounds as shown in Figure 2, it is beneficial in the long run as it leads to a faster decrease in distance to the global optimum $\mathbf{w}^*$.

**Averaging last two iterates in FedExP.** Given the oscillating behavior of the iterates of FedExP, we find that measuring progress on $F(\mathbf{w})$ using the last iterate can be misleading. Motivated by this finding, we propose to set the final model as the *average* of the last two iterates of FedExP. While the last iterate oscillates between regions that minimize the losses $F_1(\mathbf{w})$ and $F_2(\mathbf{w})$ respectively, the behavior of the average of the last two iterates is more stable and proceeds along a globally low loss region. Interestingly, we find that the benefits of averaging the iterates of FedExP also extend to training neural networks with multiple clients in practical FL scenarios (see Appendix D.1). In practice, the number of iterates to average over could also be a hyperparameter for FedExP, but we find that averaging the last two iterates works well, and we use this for our other experiments.

## 6 EXPERIMENTS

We evaluate the performance of FedExP on synthetic and real FL tasks. For our synthetic experiment, we consider a distributed overparameterized linear regression problem. This experiment aligns most closely with our theory and allows us to carefully examine the performance of FedExP when (6) holds. For realistic FL tasks, we consider image classification on the following datasets i) EMNIST (Cohen et al., 2017), ii) CIFAR-10 (Krizhevsky et al., 2009), iii) CIFAR-100 (Krizhevsky et al., 2009), iv) CINIC-10 (Darlow et al., 2018). In all experiments, we compare against the following baselines i) FedAvg, ii) SCAFFOLD (Karimireddy et al., 2020b), and iii) FedAdagrad (Reddi et al., 2021) which is a federated version of the popular Adagrad algorithm. To the best of our knowledge, we are not aware of any other baselines that adaptively tune the server step size in FL.

**Experimental Setup.** For the synthetic experiment, we consider a setup with 20 clients, 30 samples at each client, and model size to be 1000, making this an overparameterized problem. The data at each client is generated following a similar procedure as the synthetic dataset in Li et al. (2020). We use the federated version of EMNIST available at Caldas et al. (2019), which is naturally partitioned into 3400 clients. For CIFAR-10/100 we artifically partition the data into 100 clients, and for CINIC-10 we partition the data into 200 clients. In both cases, we follow a Dirichlet distribution with $\alpha = 0.3$ for the partitioning to model heterogeneity among client data (Hsu et al., 2019). For EMNIST we use the same CNN architecture used in Reddi et al. (2021). For CIFAR10, CIFAR100 and CINIC-10 we use a ResNet-18 model (He et al., 2016). For our baselines, we find the best performing $\eta_g$ and $\eta_l$ by grid-search tuning. For FedExP we optimize for $\epsilon$ and $\eta_l$ by grid search. We fix the number of participating clients to 20, minibatch size to 50 and number of local updates to 20 for all experiments. In Appendix D, we provide additional details and results, including the best performing hyperparameters, comparison with FedProx (Li et al., 2020), and results for more rounds.

**FedExP comprehensively outperforms FedAvg and baselines.** Our experimental results in Figure 3 demonstrate that FedExP clearly outperforms FedAvg and competing baselines that use the best performing $\eta_g$ and $\eta_l$ found by grid search. Moreover, FedExP does not require additional communication or storage at clients or server unlike SCAFFOLD and FedAdagrad. The order-wise improvement in the case of the convex linear regression experiment confirms our theoretical motivation for FedExP outlined in Section 3.2. In this case, since (6) is satisfied, we know that the FedExP iterates are always moving towards the optimum. For realistic FL tasks, we see a consistent

Table 1: Table showing the average number of rounds to reach desired accuracy for `FedExP` and baselines. `FedExP` provides a consistent speedup over all baselines.

| Dataset | Target Acc. | FedExP | FedAvg | SCAFFOLD | FedAdagrad |
|---------|-------------|--------|--------|----------|------------|
| EMNIST | 84% | 186 | 328 (1.76×) | 232 (1.24×) | 277 (1.48×) |
| CIFAR-10 | 72% | 267 | 434 (1.62×) | 429 (1.61×) | 419 (1.56×) |
| CIFAR-100 | 40% | 242 | 500 (2.06×) | >500 (>2.06×) | 494 (2.04×) |
| CINIC-10 | 58% | 318 | 450 (1.42×) | 470 (1.48×) | 444 (1.40×) |

speedup of over $1.4 - 2\times$ over `FedAvg`. This verifies that `FedExP` also provides performance improvement in more general settings with realistic datasets and models. Plots showing $\eta_g^{(t)}$ can be found in Appendix D.5. The key takeaway from our experiments is that adapting the server step size allows `FedExP` to take much larger steps in some (but not all) rounds compared to the constant optimum step size taken by our baselines, leading to a large speedup.

**Comparison with `FedAdagrad`.** As discussed in Section 1, `FedAdagrad` and `FedExP` use different notions of adaptivity; `FedAdagrad` uses coordinate-wise adaptivity, while `FedExP` uses client-based adaptivity. We believe that the latter is more meaningful for FL settings as seen in our experiments. In many experiments, especially image classification tasks like CIFAR, the gradients produced are *dense* with relatively little variance in coordinate-wise gradient magnitudes (Reddi et al., 2021; Zhang et al., 2020). In such cases, `FedAdagrad` is unable to leverage any coordinate-level information and gives almost the same performance as `FedAvg`.

**Comparison with `SCAFFOLD`.** We see that `FedExP` outperforms `SCAFFOLD` in all experiments, showing that adaptively tuning the server step size is sufficient to achieve speedup in FL settings. Furthermore, `SCAFFOLD` even fails to outperform `FedAvg` for the more difficult CIFAR and CINIC datasets. Several other papers have reported similar findings, including Reddi et al. (2021); Karimireddy et al. (2020a); Yu et al. (2022). Several reasons have been postulated for this behavior, including the staleness of control variates (Reddi et al., 2021) and the difficulty in characterizing client drift in non-convex scenarios (Yu et al., 2022). Thus, while theoretically attractive, simply using variance reduction techniques such as `SCAFFOLD` may not provide any speedup in practice.

**Adding extrapolation to `SCAFFOLD`.** We note that `SCAFFOLD` only modifies the Local SGD procedure at clients and keeps the global aggregation at the server unchanged. Therefore, it is easy to modify the `SCAFFOLD` algorithm to use extrapolation when updating the global model at the server (algorithm details in Appendix E). Figure 4 shows the result of our proposed extrapolated `SCAFFOLD` on the CIFAR-10 dataset. Interestingly, we observe that while `SCAFFOLD` alone fails to outperform `FedAvg`, the extrapolated version of `SCAFFOLD` achieves the best performance among all algorithms. This result highlights the importance of carefully tuning the server step size to achieve the best performance for variance-reduction algorithms. It is also possible to add extrapolation to algorithms with server momentum (Appendix F).

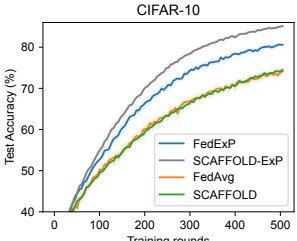

Figure 4: Adding extrapolation to `SCAFFOLD` for greater speedup.

## 7 CONCLUSION

In this paper, we have proposed `FedExP`, a novel extension of `FedAvg` that adaptively determines the server step size used in every round of global aggregation in FL. Our algorithm is based on the key observation that `FedAvg` can be seen as an approximate variant of the `POCS` algorithm, especially for overparameterized convex objectives. This has inspired us to leverage the idea of extrapolation that is used to speed up `POCS` in a federated setting, resulting in `FedExP`. We have also discussed several theoretical and empirical perspectives of `FedExP`. In particular, we have explained some design choices in `FedExP` and how it can be used in practical scenarios with partial client participation and secure aggregation. We have also shown the convergence of `FedExP` for possibly underparameterized models and non-convex objectives. Our experimental results have shown that `FedExP` consistently outperforms baseline algorithms with virtually no additional computation or communication at clients or server. We have also shown that the idea of extrapolation can be combined with other techniques, such as the variance-reduction method in `SCAFFOLD`, for greater speedup. Future work will study the convergence analysis of `FedExP` with stochastic gradient noise and the incorporation of extrapolation into a wider range of algorithms used in FL.

ACKNOWLEDGMENTS

This work was supported in part by NSF grants CCF 2045694, CNS-2112471, ONR N00014-23-1-2149, and the CMU David Barakat and LaVerne Owen-Barakat Fellowship.

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

# APPENDIX

## A ADDITIONAL RELATED WORK

In this section, we provide further discussion on some additional related work that complements our discussion in Section 1.

**Adaptive Step Size in Gradient Descent.** Here we briefly discuss methods for tuning the step size in gradient descent and the challenges in applying them to the FL setting. Early methods to tune the step size in gradient descent were based on line search (or backtracking) strategies (Armijo, 1966; Goldstein, 1977). However, these strategies need to repeatedly compute the function value or gradient within an iteration, making them computationally expensive. Another popular class of adaptive step sizes is based on the Polyak step size (Polyak, 1969; Hazan & Kakade, 2019; Loizou et al., 2021). Similar to `FedExP`, the Polyak step size is derived from trying to minimize the distance to the optimum for convex functions. However it is not clear how this can be extended to the federated setting where we only have access to pseudo-gradients. Also, the Polyak step size requires knowledge of the function value at the optimum which is hard to estimate. Another related class of step sizes is the Barzilai-Borwein stepsize (Barzilai & Borwein, 1988). However, to the best of our knowledge, these are known to provably work only for quadratic functions (Raydan, 1993; Burdakov et al., 2019) only. A recent work (Malitsky & Mishchenko, 2020) alleviates some of the concerns associated with these classical methods by setting the step size as an approximation of the inverse local Lipschitz constant; however it is again not clear how this intuition can be applied to the federated setting. An orthogonal line of work has focused on methods that adapt to the geometry of the data using gradient information in previous iterations, the most popular among them being `Adagrad` (Duchi et al., 2011) and its extensions `RMSProp` (Tieleman et al., 2012) and `Adadelta` (Zeiler, 2012). There exist federated counterparts of these algorithms, namely `FedAdagrad`; however, as we show in our experiments these methods can fail to even outperform `FedAvg` in standard FL tasks.

**Overparameterization in FL.** Inspired by the success of analyzing deep neural networks in the neural tangent kernel (NTK) regime (Jacot et al., 2018; Arora et al., 2019; Allen-Zhu et al., 2019), recent work has looked at studying the convergence of overparameterized neural networks in the FL setting. Huang et al. (2021) and Deng et al. (2022) show that for a sufficiently wide neural network and proper step size conditions, `FedAvg` will converge to a globally optimal solution even in the presence of data heterogeneity. We note that these works are primarily concerned with convergence analysis, whereas our focus is on developing a practical algorithm that is inspired by characteristics in the overparameterized regime for speeding up FL training. Another recent line of work has looked at utilizing NTK style Jacobian features for learning a FL model in just a few rounds of communication (Yu et al., 2022; Yue et al., 2022). While interesting, these approaches are orthogonal to our current work.

## B  TABLE OF NOTATION AND SCHEMATIC

### B.1  TABLE OF NOTATION

Table 2: Summary of notation used in paper

| Symbol | Description |
|---|---|
| $\|\ \|$ | $L_2$ norm |
| $M$ | Number of clients |
| $\ell(\cdot, \cdot)$ | Loss function |
| $\mathcal{D}_i$ | Dataset at $i$-th client |
| $F_i(\mathbf{w})$ | Local objective at $i$-th client |
| $F(\mathbf{w})$ | Global objective at server |
| $\eta_l$ | Client step size |
| $\eta_g$ | Server step size |
| $\mathbf{w}^{(t)}$ | Global model at round $t$ |
| $\eta_g^{(t)}$ | FedExP server step size at round $t$ |
| $\mathbf{w}_i^{(t,k)}$ | Local model at $i$-th client at $t$-th round and $k$-th iteration |
| $\tau$ | Number of local SGD steps |
| $\Delta_i^{(t)}$ | Update of $i$-th client at round $t$ |
| $\bar{\Delta}^{(t)}$ | Average of client updates at round $t$ |
| $\mathcal{S}_i^*$ | Set of minimizers of $F_i(\mathbf{w})$ |
| $T$ | Number of communication rounds |
| $\epsilon$ | Small constant added to denominator of FedExP step size |
| $\mathbf{w}^*$ | Global minimum |
| $F^*$ | Minimum value of global objective |
| $L$ | $L$-smoothness constant used in Assumption 1 |
| $\sigma_*^2$ | Upper bound on variance of client gradients at optimum (see Assumption 2) |
| $\sigma^2$ | Upper bound on variance of client gradients (see Assumption 3) |

### B.2  SCHEMATIC OF CLIENT-SERVER COMMUNICATION IN FEDEXP

At each round $t$, the server first sends global model $\mathbf{w}^{(t)}$ to all clients. Clients perform local optimization on $\mathbf{w}^{(t)}$ to compute their local models $\mathbf{w}_i^{(t,\tau)}$ and send back their update $\Delta_i^{(t)} = \mathbf{w}_i^{(t)} - \mathbf{w}_i^{(t,\tau)}$ and norm of update $\left\|\Delta_i^{(t)}\right\|^2$ to the server. This procedure is illustrated in Figure 5.

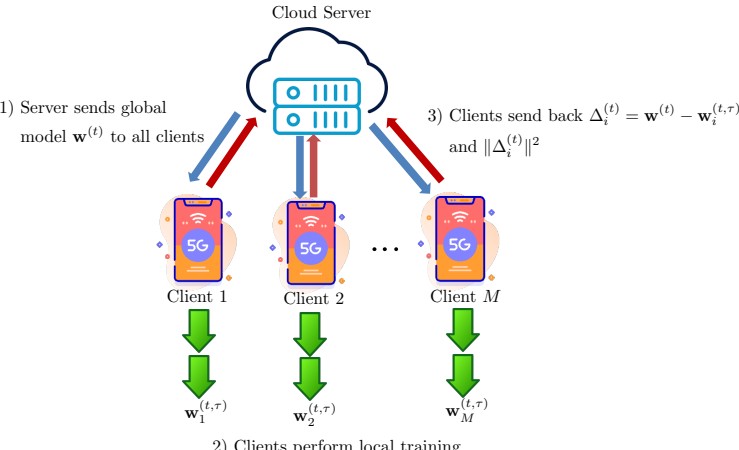

Figure 5: Schematic of client-server communication in FedExP.

## C  PROOFS

We first state some preliminary lemmas that will used throughout the proofs.

**Lemma 2.** *(Jensen's inequality) For any* $\mathbf{a}_i \in \mathbb{R}^d, i \in \{1, 2, \ldots, M\}$:

$$\left\| \frac{1}{M} \sum_{i=1}^{M} \mathbf{a}_i \right\|^2 \leq \frac{1}{M} \sum_{i=1}^{M} \|\mathbf{a}_i\|^2, \tag{13}$$

$$\left\| \sum_{i=1}^{M} \mathbf{a}_i \right\|^2 \leq M \sum_{i=1}^{M} \|\mathbf{a}_i\|^2. \tag{14}$$

We also note the following known result related to the Bregman divergence.

**Lemma 3.** *(Khaled et al., 2020) If $F$ is smooth and convex, then*

$$\|\nabla F(\mathbf{w}) - \nabla F(\mathbf{w}')\|^2 \leq 2L(F(\mathbf{w}) - F(\mathbf{w}') - \langle \nabla F(\mathbf{w}'), \mathbf{w} - \mathbf{w}' \rangle). \tag{15}$$

**Lemma 4.** *(Co-coercivity of convex smooth function) If $F$ is $L$-smooth and convex then,*

$$\langle \nabla F(\mathbf{w}) - \nabla F(\mathbf{w}'), \mathbf{w} - \mathbf{w}' \rangle \geq \frac{1}{L} \|\nabla F(\mathbf{w}) - \nabla F(\mathbf{w}')\|^2. \tag{16}$$

A direct consequence of this lemma is,

$$\langle \nabla F(\mathbf{w}), \mathbf{w} - \mathbf{w}^* \rangle \geq \frac{1}{L} \|\nabla F(\mathbf{w})\|^2 \tag{17}$$

where $\mathbf{w}^*$ is a minimizer of $F(\mathbf{w})$.

### C.1  PROOF OF LEMMA 1

Let $F_i(\mathbf{w})$ be the local objective at a client and $\mathbf{w}^*$ be the global minimum. From the overparameterization assumption, we know that $\mathbf{w}^*$ is also a minimizer for $F_i(\mathbf{w})$. We have,

$$\left\| \mathbf{w}_i^{(t,k)} - \mathbf{w}^* \right\|^2 = \left\| \mathbf{w}_i^{(t,k-1)} - \eta_l \nabla F(\mathbf{w}_i^{(t,k-1)}) - \mathbf{w}^* \right\|^2 \tag{18}$$

$$= \left\| \mathbf{w}_i^{(t,k-1)} - \mathbf{w}^* \right\|^2 - 2\eta_l \langle \nabla F(\mathbf{w}_i^{(t,k-1)}), \mathbf{w}_i^{(t,k-1)} - \mathbf{w}^* \rangle + \eta_l^2 \left\| \nabla F(\mathbf{w}_i^{(t,k-1)}) \right\|^2 \tag{19}$$

$$\leq \left\| \mathbf{w}_i^{(t,k-1)} - \mathbf{w}^* \right\|^2 - \frac{2\eta_l}{L} \left\| \nabla F(\mathbf{w}_i^{(t,k-1)}) \right\|^2 + \eta_l^2 \left\| \nabla F(\mathbf{w}_i^{(t,k-1)}) \right\|^2 \tag{20}$$

$$\leq \left\| \mathbf{w}_i^{(t,k-1)} - \mathbf{w}^* \right\|^2 - \frac{\eta_l}{L} \left\| \nabla F(\mathbf{w}_i^{(t,k-1)}) \right\|^2 \tag{21}$$

where (20) follows from (17) and (21) follows from $\eta_l \leq \frac{1}{L}$. Summing the above inequality from $k = 0$ to $\tau - 1$ we have,

$$\left\| \mathbf{w}_i^{(t,\tau)} - \mathbf{w}^* \right\|^2 \leq \left\| \mathbf{w}^{(t)} - \mathbf{w}^* \right\|^2 - \frac{\eta_l}{L} \sum_{k=0}^{\tau-1} \left\| \nabla F(\mathbf{w}_i^{(t,k)}) \right\|^2. \tag{22}$$

Thus we have,

$$\frac{1}{M} \sum_{i=1}^{M} \left\| \mathbf{w}_i^{(t,\tau)} - \mathbf{w}^* \right\|^2 \leq \left\| \mathbf{w}^{(t)} - \mathbf{w}^* \right\|^2 - \frac{\eta_l}{ML} \sum_{i=1}^{M} \sum_{k=0}^{\tau-1} \left\| \nabla F(\mathbf{w}_i^{(t,k)}) \right\|^2 \tag{23}$$

$$\leq \left\| \mathbf{w}^{(t)} - \mathbf{w}^* \right\|^2. \tag{24}$$

This completes the proof of this lemma. ☐

### C.2 Convergence Analysis for Convex Objectives

Our proof technique is inspired by Khaled et al. (2020) with some key differences. The biggest difference is the incorporation of the adaptive `FedExP` server step sizes which Khaled et al. (2020) does not account for. Another difference is that we provide convergence guarantees in terms of number of rounds $T$ while Khaled et al. (2020) focus on number of iterations $T' = T\tau$. We highlight the specific steps where we made adjustments to the analysis of Khaled et al. (2020) below.

We begin by modifying Khaled et al. (2020, Lemma 11 and Lemma 13) to bound client drift in every round instead of every iteration.

**Lemma 5.** *(Bounding client aggregate gradients)*

$$\frac{1}{M}\sum_{i=1}^{M}\sum_{k=0}^{\tau-1}\left\|\nabla F_i(\mathbf{w}_i^{(t,k)})\right\|^2 \leq \frac{3L^2}{M}\sum_{i=1}^{M}\sum_{k=0}^{\tau-1}\left\|\mathbf{w}_i^{(t,k)}-\mathbf{w}^{(t)}\right\|^2 + 6\tau L(F(\mathbf{w}^{(t)})-F(\mathbf{w}^*)) + 3\tau\sigma_*^2.$$
(25)

**Proof of Lemma 5:**

$$\frac{1}{M}\sum_{i=1}^{M}\sum_{k=0}^{\tau-1}\left\|\nabla F_i(\mathbf{w}_i^{(t,k)})\right\|^2$$

$$= \frac{1}{M}\sum_{i=1}^{M}\sum_{k=0}^{\tau-1}\left\|\nabla F_i(\mathbf{w}_i^{(t,k)}) - \nabla F_i(\mathbf{w}^{(t)}) + \nabla F_i(\mathbf{w}^{(t)}) - \nabla F_i(\mathbf{w}^*) + \nabla F_i(\mathbf{w}^*)\right\|^2 \qquad (26)$$

$$\leq \frac{3}{M}\sum_{i=1}^{M}\sum_{k=0}^{\tau-1}\left\|\nabla F_i(\mathbf{w}_i^{(t,k)}) - \nabla F_i(\mathbf{w}^{(t)})\right\|^2 + \frac{3}{M}\sum_{i=1}^{M}\sum_{k=0}^{\tau-1}\left\|\nabla F_i(\mathbf{w}^{(t)}) - \nabla F_i(\mathbf{w}^*)\right\|^2 \qquad (27)$$

$$+ \frac{3}{M}\sum_{i=1}^{M}\sum_{k=0}^{\tau-1}\left\|\nabla F_i(\mathbf{w}^*)\right\|^2$$

$$\leq \frac{3L^2}{M}\sum_{i=1}^{M}\sum_{k=0}^{\tau-1}\left\|\mathbf{w}_i^{(t,k)} - \mathbf{w}^{(t)}\right\|^2 + 6\tau L(F(\mathbf{w}^{(t)}) - F^*) + 3\tau\sigma_*^2. \qquad (28)$$

The first term in (28) follows from $L$-smoothness of $F_i(\mathbf{w})$, the second term follows from Lemma 3 and the third term follows from bounded noise at optimum. $\qquad\square$

**Lemma 6.** *(Bounding client drift)*

$$\frac{1}{M}\sum_{i=1}^{M}\sum_{k=0}^{\tau-1}\left\|\mathbf{w}^{(t)} - \mathbf{w}_i^{(t,k)}\right\|^2 \leq 12\eta_l^2\tau^2(\tau-1)L(F(\mathbf{w}^{(t)}) - F(\mathbf{w}^*)) + 6\eta_l^2\tau^2(\tau-1)\sigma_*^2. \quad (29)$$

**Proof of Lemma 6:**

$$\frac{1}{M}\sum_{i=1}^{M}\sum_{k=0}^{\tau-1}\left\|\mathbf{w}^{(t)}-\mathbf{w}_i^{(t,k)}\right\|^2$$

$$=\eta_l^2\frac{1}{M}\sum_{i=1}^{M}\sum_{k=0}^{\tau-1}\left\|\sum_{l=0}^{k-1}\nabla F_i(\mathbf{w}_i^{(t,l)})\right\|^2 \tag{30}$$

$$\leq\eta_l^2\frac{1}{M}\sum_{i=1}^{M}\sum_{k=0}^{\tau-1}k\sum_{l=0}^{k-1}\left\|\nabla F_i(\mathbf{w}_i^{(t,l)})\right\|^2 \tag{31}$$

$$\leq\eta_l^2\tau(\tau-1)\frac{1}{M}\sum_{i=1}^{M}\sum_{k=0}^{\tau-1}\left\|\nabla F_i(\mathbf{w}_i^{(t,k)})\right\|^2 \tag{32}$$

$$\leq 3\eta_l^2\tau(\tau-1)L^2\frac{1}{M}\sum_{i=1}^{M}\sum_{k=0}^{\tau-1}\left\|\mathbf{w}^{(t)}-\mathbf{w}_i^{(t,k)}\right\|^2+6\eta_l^2\tau^2(\tau-1)L(F(\mathbf{w}^{(t)})-F(\mathbf{w}^*)) \tag{33}$$

$$+3\eta_l^2\tau^2(\tau-1)\sigma_*^2$$

$$\leq\frac{1}{2M}\sum_{i=1}^{M}\sum_{k=0}^{\tau-1}\left\|\mathbf{w}^{(t)}-\mathbf{w}_i^{(t,k)}\right\|^2+6\eta_l^2\tau^2(\tau-1)L(F(\mathbf{w}^{(t)})-F(\mathbf{w}^*)) \tag{34}$$

$$+3\eta_l^2\tau^2(\tau-1)\sigma_*^2$$

where (33) uses Lemma 5 and (34) uses $\eta_l\leq\frac{1}{6\tau L}$.

Therefore we have,

$$\frac{1}{M}\sum_{i=1}^{M}\sum_{k=0}^{\tau-1}\left\|\mathbf{w}^{(t)}-\mathbf{w}_i^{(t,k)}\right\|^2\leq 12\eta_l^2\tau^2(\tau-1)L(F(\mathbf{w}^{(t)})-F(\mathbf{w}^*))+6\eta_l^2\tau^2(\tau-1)\sigma_*^2. \tag{35}$$

$$\square$$

**Proof of Theorem 1:**

We define the following auxiliary variables that will used in the proof.

$$\text{Aggregate Client Gradient:}\quad \mathbf{h}_i^{(t)}=\sum_{k=0}^{\tau-1}\nabla F_i(\mathbf{w}_i^{(t,k)}). \tag{36}$$

We also define $\bar{\mathbf{h}}^{(t)}=\frac{1}{M}\sum_{i=1}^{M}\mathbf{h}_i^{(t)}$.

Recall that the update of the global model can be written as $\mathbf{w}^{(t+1)}=\mathbf{w}^{(t)}-\eta_g^{(t)}\eta_l\bar{\mathbf{h}}^{(t)}$.

We have

$$\left\|\mathbf{w}^{(t+1)}-\mathbf{w}^*\right\|^2=\left\|\mathbf{w}^{(t)}-\eta_g^{(t)}\eta_l\bar{\mathbf{h}}^{(t)}-\mathbf{w}^*\right\|^2 \tag{37}$$

$$=\left\|\mathbf{w}^{(t)}-\mathbf{w}^*\right\|^2-2\eta_g^{(t)}\eta_l\left\langle\mathbf{w}^t-\mathbf{w}^*,\bar{\mathbf{h}}^{(t)}\right\rangle+(\eta_g^{(t)})^2\eta_l^2\left\|\bar{\mathbf{h}}^{(t)}\right\|^2 \tag{38}$$

$$\leq\left\|\mathbf{w}^{(t)}-\mathbf{w}^*\right\|^2-2\eta_g^{(t)}\eta_l\left\langle\mathbf{w}^t-\mathbf{w}^*,\bar{\mathbf{h}}^{(t)}\right\rangle+\eta_g^{(t)}\eta_l^2\frac{1}{M}\sum_{i=1}^{M}\left\|\mathbf{h}_i^{(t)}\right\|^2 \tag{39}$$

where (39) follows from $\eta_g^{(t)}\leq\frac{\sum_{i=1}^{M}\left\|\mathbf{h}_i^{(t)}\right\|^2}{M\left\|\bar{\mathbf{h}}^{(t)}\right\|^2}$. Inequality (39) is a key step in our proof and the differentiating factor in our approach from Khaled et al. (2020). Following a similar technique as Khaled et al. (2020) to bound $(\eta_g^{(t)})^2\eta_l^2\left\|\bar{\mathbf{h}}^{(t)}\right\|^2$ will end up requiring the condition $\eta_l\leq 1/8L\eta_g^{(t)}$, which cannot be satisfied in our setup due to the adaptive choice of $\eta_g^{(t)}$. Therefore we first upper

bound $(\eta_g^{(t)})^2 \eta_l^2 \left\| \bar{\mathbf{h}}^{(t)} \right\|^2$ by $\eta_g^{(t)} \eta_l^2 \frac{1}{M} \sum_{i=1}^{M} \left\| \mathbf{h}_i^{(t)} \right\|^2$ and focus on further bounding this quantity in the rest of the proof, which does not require the aforementioned condition. Note that this comes at the expense of the additional $T_3$ error seen in our final convergence bound in Theorem 1.

Therefore,

$$\left\| \mathbf{w}^{(t+1)} - \mathbf{w}^* \right\|^2 \leq \left\| \mathbf{w}^{(t)} - \mathbf{w}^* \right\|^2 - 2\eta_g^{(t)} \eta_l \underbrace{\left\langle \mathbf{w}^t - \mathbf{w}^*, \bar{\mathbf{h}}^{(t)} \right\rangle}_{T_1} + \eta_g^{(t)} \eta_l^2 \underbrace{\frac{1}{M} \sum_{i=1}^{M} \left\| \mathbf{h}_i^{(t)} \right\|^2}_{T_2}. \quad (40)$$

**Bounding $T_2$**

We have,

$$T_2 = \frac{1}{M} \sum_{i=1}^{M} \left\| \mathbf{h}_i^{(t)} \right\|^2 \quad (41)$$

$$= \frac{1}{M} \sum_{i=1}^{M} \left\| \sum_{k=0}^{\tau-1} \nabla F_i(\mathbf{w}_i^{(t,k)}) \right\|^2 \quad (42)$$

$$\leq \frac{\tau}{M} \sum_{i=1}^{M} \sum_{k=0}^{\tau-1} \left\| \nabla F_i(\mathbf{w}_i^{(t,k)}) \right\|^2 \quad (43)$$

$$\leq \frac{3\tau L^2}{M} \sum_{i=1}^{M} \sum_{k=0}^{\tau-1} \left\| \mathbf{w}_i^{(t,k)} - \mathbf{w}^{(t)} \right\|^2 + 6\tau^2 L(F(\mathbf{w}^{(t)}) - F^*) + 3\tau^2 \sigma_*^2 \quad (44)$$

where (43) follows from Jensen's inequality and and (44) follows from Lemma 5.

**Bounding $T_1$**

$$T_1 = \frac{1}{M} \sum_{i=1}^{M} \left\langle \mathbf{w}^t - \mathbf{w}^*, \mathbf{h}_i^{(t)} \right\rangle \quad (45)$$

$$= \frac{1}{M} \sum_{i=1}^{M} \sum_{k=0}^{\tau-1} \left\langle \mathbf{w}^{(t)} - \mathbf{w}^*, \nabla F_i(\mathbf{w}_i^{(t,k)}) \right\rangle. \quad (46)$$

We have,

$$\left\langle \mathbf{w}^{(t)} - \mathbf{w}^*, \nabla F_i(\mathbf{w}_i^{(t,k)}) \right\rangle = \left\langle \mathbf{w}^{(t)} - \mathbf{w}_i^{(t,k)}, \nabla F_i(\mathbf{w}_i^{(t,k)}) \right\rangle + \left\langle \mathbf{w}_i^{(t,k)} - \mathbf{w}^*, \nabla F_i(\mathbf{w}_i^{(t,k)}) \right\rangle. \quad (47)$$

From $L$-smoothness of $F_i$ we have,

$$\left\langle \mathbf{w}^{(t)} - \mathbf{w}_i^{(t,k)}, \nabla F_i(\mathbf{w}_i^{(t,k)}) \right\rangle \geq F_i(\mathbf{w}^{(t)}) - F_i(\mathbf{w}_i^{(t,k)}) - \frac{L}{2} \left\| \mathbf{w}^{(t)} - \mathbf{w}_i^{(t,k)} \right\|^2. \quad (48)$$

From convexity of $F_i$ we have,

$$\left\langle \mathbf{w}_i^{(t,k)} - \mathbf{w}^*, \nabla F_i(\mathbf{w}_i^{(t,k)}) \right\rangle \geq F_i(\mathbf{w}_i^{(t,k)}) - F_i(\mathbf{w}^*). \quad (49)$$

Therefore, adding the above inequalities we have,

$$\left\langle \mathbf{w}^{(t)} - \mathbf{w}^*, \nabla F_i(\mathbf{w}_i^{(t,k)}) \right\rangle \geq F_i(\mathbf{w}^{(t)}) - F_i(\mathbf{w}^*) - \frac{L}{2} \left\| \mathbf{w}^{(t)} - \mathbf{w}_i^{(t,k)} \right\|^2. \quad (50)$$

Substituting (50) in (46) we have,

$$T_1 \geq \tau(F(\mathbf{w}^{(t)}) - F(\mathbf{w}^*)) - \frac{L}{2M} \sum_{i=1}^{M} \sum_{k=0}^{\tau-1} \left\| \mathbf{w}^{(t)} - \mathbf{w}_i^{(t,k)} \right\|^2. \quad (51)$$

Here we would like to note that the bound for $T_1$ is our contribution and is needed in our proof due to the relaxation in (39). The bound for $T_2$ follows a similar technique as Khaled et al. (2020, Lemma 12).

Substituting the bounds for $T_1$ and $T_2$ in (40) we have,

$$
\begin{aligned}
\left\|\mathbf{w}^{(t+1)} - \mathbf{w}^*\right\|^2 &\leq \left\|\mathbf{w}^{(t)} - \mathbf{w}^*\right\|^2 - 2\eta_g^{(t)}\eta_l\tau(1 - 3\eta_l\tau L)(F(\mathbf{w}^{(t)}) - F(\mathbf{w}^*)) + 3\eta_g^{(t)}\eta_l^2\tau^2\sigma_*^2 \\
&\quad + (3\eta_g^{(t)}\eta_l^2\tau L^2 + \eta_g^{(t)}\eta_l L)\frac{1}{M}\sum_{i=1}^{M}\sum_{k=0}^{\tau-1}\left\|\mathbf{w}_i^{(t,k)} - \mathbf{w}^{(t)}\right\|^2 \\
&\leq \left\|\mathbf{w}^{(t)} - \mathbf{w}^*\right\|^2 - \eta_g^{(t)}\eta_l\tau(F(\mathbf{w}^{(t)}) - F(\mathbf{w}^*)) + 3\eta_g^{(t)}\eta_l^2\tau^2\sigma_*^2 \qquad (52) \\
&\quad + 2\eta_g^{(t)}\eta_l L\frac{1}{M}\sum_{i=1}^{M}\sum_{k=0}^{\tau-1}\left\|\mathbf{w}_i^{(t,k)} - \mathbf{w}^{(t)}\right\|^2 \\
&\leq \left\|\mathbf{w}^{(t)} - \mathbf{w}^*\right\|^2 - \eta_g^{(t)}\eta_l\tau(F(\mathbf{w}^{(t)}) - F(\mathbf{w}^*)) + 3\eta_g^{(t)}\eta_l^2\tau^2\sigma_*^2 \qquad (53) \\
&\quad + 24\eta_g^{(t)}\eta_l^3\tau^2(\tau - 1)L^2(F(\mathbf{w}^{(t)}) - F(\mathbf{w}^*)) + 12\eta_g^{(t)}\eta_l^3\tau^2(\tau - 1)L\sigma_*^2 \\
&\leq \left\|\mathbf{w}^{(t)} - \mathbf{w}^*\right\|^2 - \frac{\eta_g^{(t)}\eta_l\tau}{3}(F(\mathbf{w}^{(t)}) - F(\mathbf{w}^*)) + 3\eta_g^{(t)}\eta_l^2\tau^2\sigma_*^2 \qquad (54) \\
&\quad + 12\eta_g^{(t)}\eta_l^3\tau^2(\tau - 1)L\sigma_*^2
\end{aligned}
$$

where both (52) and (55) use $\eta_l \leq \frac{1}{6\tau L}$, and (53) uses Lemma 6.

Rearranging terms and averaging over all rounds we have,

$$
\frac{\sum_{t=0}^{T-1}\eta_g^{(t)}F(\mathbf{w}^{(t)}) - F(\mathbf{w}^*)}{\sum_{t=0}^{T-1}\eta_g^{(t)}} \leq \frac{3\left\|\mathbf{w}^{(0)} - \mathbf{w}^*\right\|^2}{\sum_{t=0}^{T-1}\eta_g^{(t)}\eta_l\tau} + 9\eta_l\tau\sigma_*^2 + 36\eta_l^2\tau(\tau - 1)L\sigma_*^2. \qquad (55)
$$

This implies,

$$
F(\bar{\mathbf{w}}^{(T)}) - F(\mathbf{w}^*) \leq \mathcal{O}\left(\frac{\left\|\mathbf{w}^{(0)} - \mathbf{w}^*\right\|^2}{\eta_l\tau\sum_{t=0}^{T-1}\eta_g^{(t)}}\right) + \mathcal{O}\left(\eta_l^2\tau(\tau - 1)L\sigma_*^2\right) + \mathcal{O}\left(\eta_l\tau\sigma_*^2\right) \qquad (56)
$$

where $\bar{\mathbf{w}}^{(T)} = \frac{\sum_{t=0}^{T-1}\eta_g^{(t)}\mathbf{w}^{(t)}}{\sum_{t=0}^{T-1}\eta_g^{(t)}}$. This completes the proof. $\qquad\square$

## C.3 Convergence Analysis for Non-Convex Objectives

Our proof technique is inspired by Wang et al. (2020) and we use one of their intermediate results to bound client drift in non-convex settings as we describe below. We highlight the specific steps where we made adjustments to the analysis of Wang et al. (2020) below.

We begin by defining the following auxiliary variables that will used in the proof.

$$
\text{Normalized Gradient:} \quad \mathbf{h}_i^{(t)} = \frac{1}{\tau}\sum_{k=0}^{\tau-1}\nabla F_i(\mathbf{w}_i^{(t,k)}). \qquad (57)
$$

We also define $\bar{\mathbf{h}}^{(t)} = \frac{1}{M}\sum_{i=1}^{M}\mathbf{h}_i^{(t)}$.

**Lemma 7.** *(Bounding client drift in Non-Convex Setting)*

$$
\frac{1}{M}\sum_{i=1}^{M}\left\|\nabla F_i(\mathbf{w}^{(t)}) - \mathbf{h}_i^{(t)}\right\|^2 \leq \frac{1}{8}\left\|\nabla F(\mathbf{w}^{(t)})\right\|^2 + 5\eta_l^2 L^2\tau(\tau - 1)\sigma_g^2. \qquad (58)
$$

**Proof of Lemma 7:** Let $D = 4\eta_l^2 L^2 \tau(\tau - 1)$. We have the following bound from equation (87) in Wang et al. (2020),

$$\frac{1}{M} \sum_{i=1}^{M} \left\| \nabla F_i(\mathbf{w}^{(t)}) - \mathbf{h}_i^{(t)} \right\|^2 \leq \frac{D}{1 - D} \left\| \nabla F(\mathbf{w}^{(t)}) \right\|^2 + \frac{D\sigma_g^2}{1 - D}. \tag{59}$$

From $\eta_l \leq \frac{1}{6\tau L}$ we have $D \leq \frac{1}{9}$ which implies $\frac{1}{1-D} \leq \frac{9}{8}$ and $\frac{D}{1-D} \leq \frac{1}{8}$.

Therefore we have,

$$\frac{1}{M} \sum_{i=1}^{M} \left\| \nabla F_i(\mathbf{w}^{(t)}) - \mathbf{h}_i^{(t)} \right\|^2 \leq \frac{1}{8} \left\| \nabla F(\mathbf{w}^{(t)}) \right\|^2 + \frac{9D}{8}\sigma_g^2 \tag{60}$$

$$\leq \frac{1}{8} \left\| \nabla F(\mathbf{w}^{(t)}) \right\|^2 + 5\eta_l^2 L^2 \tau(\tau - 1)\sigma_g^2. \tag{61}$$

$\square$

**Proof of Theorem 2:**

The update of the global model can be written as follows,

$$\mathbf{w}^{(t+1)} = \mathbf{w}^{(t)} - \eta_g^{(t)} \eta_l \tau \bar{\mathbf{h}}^{(t)}. \tag{62}$$

Now using the Lipschitz-smoothness assumption we have,

$$F(\mathbf{w}^{(t+1)}) - F(\mathbf{w}^{(t)}) \leq -\eta_g^{(t)} \eta_l \tau \left\langle \nabla F(\mathbf{w}^{(t)}), \bar{\mathbf{h}}^{(t)} \right\rangle + \frac{(\eta_g^{(t)})^2 \eta_l^2 \tau^2 L}{2} \left\| \bar{\mathbf{h}}^{(t)} \right\|^2 \tag{63}$$

$$\leq -\eta_g^{(t)} \eta_l \tau \left\langle \nabla F(\mathbf{w}^{(t)}), \bar{\mathbf{h}}^{(t)} \right\rangle + \frac{\eta_g^{(t)} \eta_l^2 \tau^2 L}{2M} \sum_{i=1}^{M} \left\| \mathbf{h}_i^{(t)} \right\|^2 \tag{64}$$

where (64) uses $\eta_g^{(t)} \leq \frac{\sum_{i=1}^{M} \left\| \mathbf{h}_i^{(t)} \right\|^2}{M \left\| \bar{\mathbf{h}}^{(t)} \right\|^2}$. As in the convex case, inequality (64) is a key step in our proof and the differentiating factor in our approach from Wang et al. (2020). Following a similar technique as Wang et al. (2020) to bound $(\eta_g^{(t)})^2 \eta_l^2 \tau^2 L \left\| \bar{\mathbf{h}}^{(t)} \right\|^2 / 2$ will need the condition $\eta_l \leq 1/2L\tau\eta_g^{(t)}$, which cannot be satisfied in our setup due to the adaptive choice of $\eta_g^{(t)}$. Therefore we first upper bound $(\eta_g^{(t)})^2 \eta_l^2 \tau^2 L^2 \left\| \bar{\mathbf{h}}^{(t)} \right\|^2$ by $\eta_g^{(t)} \eta_l^2 \tau^2 L \frac{1}{M} \sum_{i=1}^{M} \left\| \mathbf{h}_i^{(t)} \right\|^2 / 2$ and focus on further bounding this quantity in the rest of the proof, which does not require the aforementioned condition. Note that this comes at the expense of the additional $T_3$ error seen in our final convergence bound in Theorem 2.

Therefore we have,

$$F(\mathbf{w}^{(t+1)}) - F(\mathbf{w}^{(t)}) \leq \underbrace{-\eta_g^{(t)} \eta_l \tau \left\langle \nabla F(\mathbf{w}^{(t)}), \bar{\mathbf{h}}^{(t)} \right\rangle}_{T_1} + \underbrace{\frac{\eta_g^{(t)} \eta_l^2 \tau^2 L}{2M} \sum_{i=1}^{M} \left\| \mathbf{h}_i^{(t)} \right\|^2}_{T_2}. \tag{65}$$

**Bounding $T_1$**

We have,

$$T_1 = \left\langle \nabla F(\mathbf{w}^{(t)}), \frac{1}{M} \sum_{i=0}^{M} \mathbf{h}_i^{(t)} \right\rangle \tag{66}$$

$$= \frac{1}{2} \left\| \nabla F(\mathbf{w}^{(t)}) \right\|^2 + \frac{1}{2} \left\| \frac{1}{M} \sum_{i=1}^{M} \mathbf{h}_i^{(t)} \right\|^2 - \frac{1}{2} \left\| \nabla F(\mathbf{w}^{(t)}) - \frac{1}{M} \sum_{i=1}^{M} \mathbf{h}_i^{(t)} \right\|^2 \tag{67}$$

$$\geq \frac{1}{2} \left\| \nabla F(\mathbf{w}^{(t)}) \right\|^2 - \frac{1}{2M} \sum_{i=1}^{M} \left\| \nabla F_i(\mathbf{w}^{(t)}) - \mathbf{h}_i^{(t)} \right\|^2 \tag{68}$$

where (67) uses $\langle \mathbf{a}, \mathbf{b} \rangle = \frac{1}{2} \|\mathbf{a}\|^2 + \frac{1}{2} \|\mathbf{b}\|^2 - \frac{1}{2} \|\mathbf{a} - \mathbf{b}\|^2$ and (68) uses Jensen's inequality and the definition of the global objective function $F$.

**Bounding $T_2$**

We have,

$$T_2 = \frac{1}{M} \sum_{i=1}^{M} \left\| \mathbf{h}_i^{(t)} \right\|^2 \tag{69}$$

$$= \frac{1}{M} \sum_{i=1}^{M} \left\| \mathbf{h}_i^{(t)} - \nabla F_i(\mathbf{w}^{(t)}) + \nabla F_i(\mathbf{w}^{(t)}) - \nabla F(\mathbf{w}^{(t)}) + \nabla F(\mathbf{w}^{(t)}) \right\|^2 \tag{70}$$

$$\leq \frac{3}{M} \sum_{i=1}^{M} \left( \left\| \mathbf{h}_i^{(t)} - \nabla F_i(\mathbf{w}^{(t)}) \right\|^2 + \left\| \nabla F_i(\mathbf{w}^{(t)}) - \nabla F(\mathbf{w}^{(t)}) \right\|^2 + \left\| \nabla F(\mathbf{w}^{(t)}) \right\|^2 \right) \tag{71}$$

$$\leq \frac{3}{M} \sum_{i=1}^{M} \left\| \mathbf{h}_i^{(t)} - \nabla F_i(\mathbf{w}^{(t)}) \right\|^2 + 3\sigma_g^2 + 3 \left\| \nabla F(\mathbf{w}^{(t)}) \right\|^2 \tag{72}$$

where (71) uses Jensen's inequality, (72) uses bounded data heterogeneity assumption.

Here we would like to note that the bound for $T_2$ is our contribution and is needed in our proof due to the relaxation in (39). The bound for $T_1$ follows a similar technique as in Wang et al. (2020).

Substituting the $T_1$ and $T_2$ bounds into (65), we have,

$$F(\mathbf{w}^{(t+1)}) - F(\mathbf{w}^{(t)}) \leq -\eta_g^{(t)} \eta_l \tau \left( \frac{1}{2} \left\| \nabla F(\mathbf{w}^{(t)}) \right\|^2 + \frac{1}{2M} \sum_{i=1}^{M} \left\| \nabla F_i(\mathbf{w}^{(t)}) - \mathbf{h}_i^{(t)} \right\|^2 \right. \tag{73}$$

$$\left. + \frac{\eta_l \tau L}{2} \left( 3\sigma_g^2 + 3 \left\| \nabla F(\mathbf{w}^{(t)}) \right\|^2 + \frac{3}{M} \sum_{i=1}^{M} \left\| \mathbf{h}_i^{(t)} - \nabla F_i(\mathbf{w}^{(t)}) \right\|^2 \right) \right)$$

$$\leq -\eta_g^{(t)} \eta_l \tau \left( \frac{1}{4} \left\| \nabla F(\mathbf{w}^{(t)}) \right\|^2 + \frac{1}{M} \sum_{i=1}^{M} \left\| \nabla F_i(\mathbf{w}^{(t)}) - \mathbf{h}_i^{(t)} \right\|^2 + 3\eta_l \tau L \sigma_g^2 \right) \tag{74}$$

$$\leq -\eta_g^{(t)} \eta_l \tau \left( \frac{1}{8} \left\| \nabla F(\mathbf{w}^{(t)}) \right\|^2 + 3\eta_l \tau L \sigma_g^2 + 5\eta_l^2 L^2 \tau (\tau - 1) \sigma_g^2 \right) \tag{75}$$

where (74) uses $\eta_l \leq \frac{1}{6\tau L}$, (75) uses Lemma 7.

Thus rearranging terms and averaging over all rounds we have,

$$\frac{\sum_{t=0}^{T-1} \eta_g^{(t)} \left\| \nabla F(\mathbf{w}^{(t)}) \right\|^2}{\sum_{t=0}^{T-1} \eta_g^{(t)}} \leq \frac{8(F(\mathbf{w}^{(0)}) - F^*)}{\sum_{t=0}^{T-1} \eta_g^{(t)} \eta_l \tau} + 40\eta_l^2 L^2 \tau (\tau - 1) \sigma_g^2 + 24\eta_l L \tau \sigma_g^2. \tag{76}$$

This implies,

$$\min_{t \in [T]} \left\| \nabla F(\mathbf{w}^{(t)}) \right\|^2 \leq \mathcal{O} \left( \frac{(F(\mathbf{w}^{(0)}) - F^*)}{\sum_{t=0}^{T-1} \eta_g^{(t)} \eta_l \tau} \right) + \mathcal{O} \left( \eta_l^2 L^2 \tau (\tau - 1) \sigma_g^2 \right) + \mathcal{O} \left( \eta_l L \tau \sigma_g^2 \right). \tag{77}$$

This completes the proof. $\qquad \square$

## C.4 Exact Projection with Gradient Descent for Linear Regression

Let $F(\mathbf{w}) = \|\mathbf{A}\mathbf{w} - \mathbf{b}\|^2$ where $\mathbf{A}$ is a $(n \times d)$ matrix and $\mathbf{b}$ is a $n$ dimensional vector. We assume that $d \geq n$ here and $\mathbf{A}$ has rank $n$. The singular value decomposition (SVD) of $\mathbf{A}$ can be written as,

$$\mathbf{A} = \mathbf{U}\boldsymbol{\Sigma}\mathbf{V}^\top = \mathbf{U} \begin{bmatrix} \boldsymbol{\Sigma}_1 & \mathbf{0} \end{bmatrix} \begin{bmatrix} \mathbf{V}_1^\top \\ \mathbf{V}_2^\top \end{bmatrix} = \mathbf{U}\boldsymbol{\Sigma}_1\mathbf{V}_1^\top \tag{78}$$

where $\mathbf{U}$ is an $(n \times n)$ orthogonal matrix, $\mathbf{\Sigma}$ is an $(n \times n)$ diagonal matrix, $\mathbf{V}_1$ is a $(d \times n)$ matrix with orthogonal columns and $\mathbf{V}_2$ is a $(d \times (d-n))$ matrix with orthogonal columns. Here $\mathbf{V}_1$ is a basis for the row space of $\mathbf{A}$, while $\mathbf{V}_2$ is a basis for the null space of $\mathbf{A}$. We first prove the following lemmas about the set of minimizers of $F(\mathbf{w})$ and the projection on this set.

**Lemma 8.** *The set of minimizers of $F(\mathbf{w})$ is given by,*

$$\mathcal{S}^* = \{\mathbf{V}_2\mathbf{V}_2^\top\mathbf{w} + \mathbf{V}_1\mathbf{\Sigma}_1^{-1}\mathbf{U}^\top\mathbf{b}|\mathbf{w} \in \mathbb{R}^d\}. \tag{79}$$

**Proof.** Let $\mathbf{w} = \mathbf{V}_2\mathbf{V}_2^\top\mathbf{x} + \mathbf{V}_1\mathbf{\Sigma}_1^{-1}\mathbf{U}^\top\mathbf{b}$ for some $\mathbf{x} \in \mathbb{R}^d$. We have,

$$\mathbf{A}\mathbf{w} = \mathbf{U}\mathbf{\Sigma}_1\mathbf{V}_1^\top(\mathbf{V}_2\mathbf{V}_2^\top\mathbf{x} + \mathbf{V}_1\mathbf{\Sigma}_1^{-1}\mathbf{U}^\top\mathbf{b}) \tag{80}$$

$$= \mathbf{b} \tag{81}$$

where the last line uses $\mathbf{V}_1^\top\mathbf{V}_2 = 0, \mathbf{V}_1^\top\mathbf{V}_1 = \mathbf{I}, \mathbf{U}\mathbf{U}^\top = \mathbf{I}$. This implies $\|\mathbf{A}\mathbf{w} - \mathbf{b}\|^2 = 0$. Thus any $\mathbf{w}$ in $\mathcal{S}^*$ is a minimizer of $F(\mathbf{w})$.

Now let $\mathbf{w}^*$ be a minimizer of $F(\mathbf{w})$, implying $\mathbf{A}\mathbf{w}^* = \mathbf{U}\mathbf{\Sigma}_1\mathbf{V}_1^\top\mathbf{w}^* = \mathbf{b}$. We have,

$$\mathbf{w}^* = \mathbf{V}_2\mathbf{V}_2^\top\mathbf{w}^* + \mathbf{V}_1\mathbf{V}_1^\top\mathbf{w}^* \tag{82}$$

$$= \mathbf{V}_2\mathbf{V}_2^\top\mathbf{w}^* + \mathbf{V}_1\mathbf{\Sigma}_1^{-1}\mathbf{U}^\top\mathbf{b} \tag{83}$$

where (82) uses $\mathbf{V}_1\mathbf{V}_1^\top + \mathbf{V}_2\mathbf{V}_2^\top = \mathbf{I}$ and (83) uses $\mathbf{U}\mathbf{\Sigma}_1\mathbf{V}_1^\top\mathbf{w}^* = \mathbf{b}$. Thus any minimizer of $F(\mathbf{w})$ must lie in $\mathcal{S}^*$.

Combining the above statements we have,

$$\mathbf{w} \text{ is a minimizer of } F(\mathbf{w}) \iff \mathbf{w} \in \mathcal{S}^*. \tag{84}$$

which completes the proof. $\qquad\square$

**Lemma 9.** *The projection of any $\mathbf{w} \in \mathbb{R}^d$ on $\mathcal{S}^*$ is given by,*

$$P_{\mathcal{S}^*}(\mathbf{w}) = \underset{\mathbf{w}' \in \mathcal{S}^*}{\arg\min} \|\mathbf{w} - \mathbf{w}'\|^2 = \mathbf{V}_2\mathbf{V}_2^\top\mathbf{w} + \mathbf{V}_1\mathbf{\Sigma}_1^{-1}\mathbf{U}^\top\mathbf{b}. \tag{85}$$

**Proof.** When $\mathbf{w} \in \mathcal{S}^*$, it is easy to see that this holds. Therefore we consider the case where $\mathbf{w} \notin \mathcal{S}^*$. Let $\mathbf{x} = \mathbf{V}_2\mathbf{V}_2^\top\mathbf{w} + \mathbf{V}_1\mathbf{\Sigma}_1^{-1}\mathbf{U}^\top\mathbf{b}$ and $P_{\mathcal{S}^*}(\mathbf{w}) = \mathbf{V}_2\mathbf{V}_2^\top\mathbf{w}_0 + \mathbf{V}_1\mathbf{\Sigma}_1^{-1}\mathbf{U}^\top\mathbf{b}$ where $\mathbf{w}_0 \neq \mathbf{w}$. We have,

$$\left\|\mathbf{w} - \mathbf{V}_2\mathbf{V}_2^\top\mathbf{w}_0 - \mathbf{V}_1\mathbf{\Sigma}_1^{-1}\mathbf{U}^\top\mathbf{b}\right\|^2 \tag{86}$$

$$= \left\|\mathbf{V}_2\mathbf{V}_2^\top(\mathbf{w} - \mathbf{w}_0) + \mathbf{V}_1\mathbf{V}_1^\top\mathbf{w} - \mathbf{V}_1\mathbf{\Sigma}_1^{-1}\mathbf{U}^\top\mathbf{b}\right\|^2 \quad (\mathbf{V}_1\mathbf{V}_1^\top + \mathbf{V}_2\mathbf{V}_2^\top = \mathbf{I}) \tag{87}$$

$$= \left\|\mathbf{V}_2\mathbf{V}_2^\top(\mathbf{w} - \mathbf{w}_0)\right\|^2 + \left\|\mathbf{V}_1\mathbf{V}_1^\top\mathbf{w} - \mathbf{V}_1\mathbf{\Sigma}_1^{-1}\mathbf{U}^\top\mathbf{b}\right\|^2 \tag{88}$$

$$= \left\|\mathbf{V}_2\mathbf{V}_2^\top(\mathbf{w} - \mathbf{w}_0)\right\|^2 + \|\mathbf{w} - \mathbf{x}\|^2 \tag{89}$$

$$> \|\mathbf{w} - \mathbf{x}\|^2 \tag{90}$$

leading to a contradiction. The cross term in (88) is zero since $\mathbf{V}_1^\top\mathbf{V}_2 = \mathbf{0}$. Equation (89) follows by the definition of $\mathbf{x}$. $\qquad\square$

We now show that running gradient descent on $F(\mathbf{w})$ starting from $\mathbf{w}$ with a sufficiently small step size converges to $P_{\mathcal{S}^*}(\mathbf{w})$.

**Lemma 10.** *Let $\mathbf{w}^{(0)}, \mathbf{w}^{(1)}, \ldots$ be the iterates generated by running gradient descent on $F(\mathbf{w})$ with $\mathbf{w}^{(0)} = \mathbf{w}$ and learning rate $\eta_l \leq \lambda_{\max}$, where $\lambda_{\max}$ is the largest eigen value of $\mathbf{A}^\top\mathbf{A}$. Then $\lim_{T\to\infty} \mathbf{w}^{(T)} = P_{\mathcal{S}^*}(\mathbf{w})$.*

**Proof.** By the gradient descent update we have,

$$\mathbf{w}^{(t+1)} = \mathbf{w}^{(t)} - \eta_l(\mathbf{A}^\top\mathbf{A}\mathbf{w}^{(t)} - \mathbf{A}^\top\mathbf{b}) \tag{91}$$

$$= (\mathbf{I} - \eta_l\mathbf{A}^\top\mathbf{A})\mathbf{w}^{(t)} + \eta_l\mathbf{A}^\top\mathbf{b}. \tag{92}$$

Therefore,

$$\mathbf{w}^{(T)} = (\mathbf{I} - \eta_l \mathbf{A}^\top \mathbf{A})^T \mathbf{w}^{(0)} + \eta_l \sum_{t=0}^{T-1} (\mathbf{I} - \eta_l \mathbf{A}^\top \mathbf{A})^t \mathbf{A}^\top \mathbf{b} \tag{93}$$

$$= \mathbf{V}(\mathbf{I} - \eta_l \mathbf{\Sigma}^\top \mathbf{\Sigma})^T \mathbf{V}^\top \mathbf{w}^{(0)} + \eta_l \sum_{t=0}^{T-1} \mathbf{V}(\mathbf{I} - \eta_l \mathbf{\Sigma}^\top \mathbf{\Sigma})^t \mathbf{\Sigma}^\top \mathbf{U}^\top \mathbf{b} \tag{94}$$

$$= (\mathbf{V}_1 (\mathbf{I} - \eta_l \mathbf{\Sigma}_1^2)^T \mathbf{V}_1 + \mathbf{V}_2 \mathbf{V}_2^\top) \mathbf{w}^{(0)} + \eta_l \mathbf{V}_1 \left( \sum_{t=0}^{T-1} (\mathbf{I} - \eta_l \mathbf{\Sigma}_1^2)^t \right) \mathbf{\Sigma}_1 \mathbf{U}^\top \mathbf{b}. \tag{95}$$

In the limit $T \to \infty$ and with $\eta_l \leq \lambda_{\max}$, we have,

$$\lim_{T \to \infty} (\mathbf{I} - \eta_l \mathbf{\Sigma}_1^2)^T = \mathbf{0} \text{ and } \lim_{T \to \infty} \sum_{t=0}^{T-1} (\mathbf{I} - \eta_l \mathbf{\Sigma}_1^2)^t = \frac{1}{\eta_l} \mathbf{\Sigma}_1^{-2}. \tag{96}$$

Thus,

$$\lim_{T \to \infty} \mathbf{w}^{(T)} = \mathbf{V}_2 \mathbf{V}_2^\top \mathbf{w}^{(0)} + \mathbf{V}_1 \mathbf{\Sigma}_1^{-1} \mathbf{U}^\top \mathbf{b} \tag{97}$$

$$= P_{\mathcal{S}^*}(\mathbf{w}^{(0)}) \tag{98}$$

$$= P_{\mathcal{S}^*}(\mathbf{w}). \tag{99}$$

$\square$

### C.4.1 IMPROVING LOWER BOUND IN (8) IN THE CASE OF EXACT PROJECTIONS

Let $\mathcal{S}_i^*$ be convex and let $\mathbf{w}^* \in \mathcal{S}_i$ for all $i \in [M]$. We assume that $\mathbf{w}_i^{(t,\tau)} = P_{\mathcal{S}_i^*}(\mathbf{w}^{(t)}) \ \forall i \in [M]$, i.e., the local models are an exact projection of $\mathbf{w}^{(t)}$ on their respective solution sets. From (8) we have,

$$(\eta_g^{(t)})_{\text{opt}} = \frac{\left\langle \mathbf{w}^{(t)} - \mathbf{w}^*, \bar{\Delta}^{(t)} \right\rangle}{\left\| \bar{\Delta}^{(t)} \right\|^2} = \frac{\sum_{i=1}^M \left\langle \mathbf{w}^{(t)} - \mathbf{w}^*, \Delta_i^{(t)} \right\rangle}{M \left\| \bar{\Delta}^{(t)} \right\|^2}. \tag{100}$$

We can lower bound $\left\langle \mathbf{w}^{(t)} - \mathbf{w}^*, \Delta_i^{(t)} \right\rangle$ as follows,

$$\left\langle \mathbf{w}^{(t)} - \mathbf{w}^*, \Delta_i^{(t)} \right\rangle = \left\langle \mathbf{w}^{(t)} - \mathbf{w}_i^{(t,\tau)} + \mathbf{w}_i^{(t,\tau)} - \mathbf{w}^*, \mathbf{w}^{(t)} - \mathbf{w}_i^{(t,\tau)} \right\rangle \tag{101}$$

$$= \left\| \mathbf{w}^{(t)} - \mathbf{w}_i^{(t,\tau)} \right\|^2 + \left\langle \mathbf{w}_i^{(t,\tau)} - \mathbf{w}^*, \mathbf{w}^{(t)} - \mathbf{w}_i^{(t,\tau)} \right\rangle \tag{102}$$

$$\geq \left\| \mathbf{w}^{(t)} - \mathbf{w}_i^{(t,\tau)} \right\|^2 \tag{103}$$

$$= \left\| \Delta_i^{(t)} \right\|^2 \tag{104}$$

$$\tag{105}$$

where (103) uses the fact that $\left\langle \mathbf{w}_i^{(t,\tau)} - \mathbf{w}^*, \mathbf{w}^{(t)} - \mathbf{w}_i^{(t,\tau)} \right\rangle \geq 0$ following the properties of projection (Boyd & Dattarro, 2003).

Thus we have,

$$(\eta_g^{(t)})_{\text{opt}} \geq \frac{\sum_{i=1}^M \left\| \Delta_i^{(t)} \right\|^2}{M \left\| \bar{\Delta}^{(t)} \right\|^2} \tag{106}$$

Note here the improvement by a factor of 2 in the lower bound compared to (8).

# D    ADDITIONAL EXPERIMENTS AND SETUP DETAILS

Our code is available at the following link `https://github.com/Divyansh03/FedExP`.

## D.1    IMPACT OF AVERAGING ITERATES FOR NEURAL NETWORKS

As discussed in Section 5, we find that setting the final `FedExP` model as the average of the last two iterates also improves performance when training neural networks in practical FL scenarios. To demonstrate this, we consider an experiment on the CIFAR-10 dataset with 10 clients, where the data at each client is distributed using a Dirichlet distribution with $\alpha = 0.3$. We set the number of local steps to be $\tau = 20$ and train a CNN model having the same architecture as outlined in McMahan et al. (2017) with full client participation. Figure 6 shows the training accuracy as a function of the last iterate and the average of last two iterates for `FedAvg` and `FedExP`. We see that the last iterate of `FedExP` has an oscillating behavior that can hide improvements in training accuracy. On the other hand, the average of the last two iterates of `FedExP` produces a more stable training curve and shows a considerable improvement in the final accuracy. Note however that this improvement only shows for `FedExP`; averaging iterates does not make significant difference for `FedAvg`.

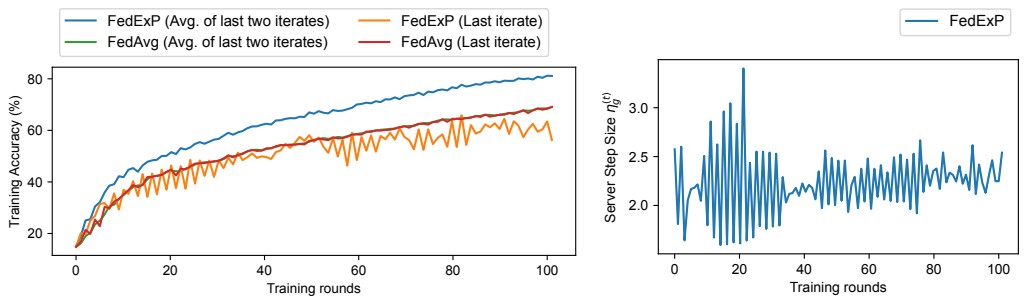

Figure 6:  Benefit of averaging the last two iterates for `FedExP` in training a CNN model on CIFAR-10. Note that averaging does not make significant difference for `FedAvg`.

## D.2    DATASET DETAILS

Here we provide more details about the datasets used in Section 6.

**Synthetic Linear Regression.**    In this case we assume that the local objective of each client is given by $F_i(\mathbf{w}) = \|\mathbf{A}_i \mathbf{w} - \mathbf{b}_i\|^2$ where $\mathbf{A}_i \in \mathbb{R}^{(30 \times 1000)}$, $\mathbf{b}_i \in \mathbb{R}^{30}$ and $\mathbf{w} \in \mathbb{R}^{1000}$. We set the number of clients to be $M = 20$. Note that since $d \geq \sum_{i=1}^{M} n_i$, this is an overparameterized convex problem. To generate $\mathbf{A}_i$ and $\mathbf{b}_i$, we follow a similar process as Li et al. (2020). We have $(\mathbf{A}_i)_{j:} \sim \mathcal{N}(\mathbf{m}_i, \mathbf{I}_d)$ and $(\mathbf{b}_i)_j = \mathbf{w}_i^\top (\mathbf{A}_i)_{j:}$ where $\mathbf{m}_i \sim \mathcal{N}(u_i, 1), \mathbf{w}_i \sim \mathcal{N}(y_i, 1), u_i \sim \mathcal{N}(0, 0.1), y_i \sim \mathcal{N}(0, 0.1)$.

**EMNIST.**    EMNIST is an image classification task consisting of handwritten characters associated with 62 labels. The federated EMNIST dataset available at Caldas et al. (2019) is naturally partitioned into 3400 clients based on the identities of the character authors. The number of training and test samples is 671,585 and 77,483 respectively.

**CIFAR-10/100.**    CIFAR-10 is a natural image dataset consisting of 60,000 32x32 images divided into 10 classes. CIFAR-100 uses a finer labeling of the CIFAR images to divide them into 100 classes making it a harder dataset for image classification. In both cases the number of training examples and test examples is 50,000 and 10,000 respectively. To simulate a federated setting, we artificially partition the training data into 100 clients following the procedure outlined in Hsu et al. (2019).

**CINIC-10.**    CINIC-10 is a natural image dataset that can be used as a direct replacement of CIFAR for machine learning tasks. It is intended to act as a harder dataset than CIFAR-10 while being easier than CIFAR-100. The number of training and test examples is both 90,000. We partition the training data into 200 clients in this case, following a similar procedure as for CIFAR.

## D.3 HYPERPARAMETER DETAILS

For our baselines, we find the best performing $\eta_g$ and $\eta_l$ by grid-search tuning. For FedExP we search for $\epsilon$ and $\eta_l$. This is done by running algorithms for 50 rounds and finding the parameters that achieve the highest training accuracy averaged over the last 10 rounds. We provide details of the grid used below for each experiment below.

**Grid for Synthetic.**
For FedAvg and SCAFFOLD, the grid for $\eta_g$ is $\{10^0, 10^{0.5}, 10^{0.5}, 10^1, 10^2\}$. For FedAdagrad, the grid for $\eta_g$ is $\{10^{-1}, 10^{-0.5}, 10^{-0}, 10^{0.5}, 10^1\}$. For FedExP we keep $\epsilon = 0$ in this experiment as (6) is satisfied in this case. The grid for $\eta_l$ is $\{10^{-2}, 10^{-1.5}, 10^{-1}, 10^{-0.5}, 10^0\}$ for all algorithms.

**Grid for Neural Network Experiments.**
For FedAvg and SCAFFOLD the grid for $\eta_g$ is $\{10^{-1}, 10^{-0.5}, 10^0, 10^{0.5}, 10^1\}$. For FedAdagrad, the grid for $\eta_g$ is $\{10^{-2}, 10^{-1.5}, 10^{-1}, 10^{-0.5}, 10^0\}$. For FedExP the grid for $\epsilon$ is $\{10^{-3}, 10^{-2.5}, 10^{-2}, 10^{-1.5}, 10^{-1}\}$. The grid for $\eta_l$ is $\{10^{-2}, 10^{-1.5}, 10^{-1}, 10^{-0.5}, 10^0\}$ for all algorithms.

We use lower values of $\eta_g$ in the grid for FedAdagrad based on observations from Reddi et al. (2021) which show that FedAdagrad performs better with smaller values of the server step size. We provide details of the best performing hyperparameters below.

Table 3: Base-10 logarithm of the best combination of $\epsilon$ and $\eta_l$ for FedExP and combination of $\eta_l$ and $\eta_g$ for baselines. For the synthetic dataset we keep $\epsilon = 0$ for FedExP.

| Dataset | FedExP | | FedAvg | | SCAFFOLD | | FedAdagrad | |
|---|---|---|---|---|---|---|---|---|
| | $\epsilon$ | $\eta_l$ | $\eta_g$ | $\eta_l$ | $\eta_g$ | $\eta_l$ | $\eta_g$ | $\eta_l$ |
| Synthetic | * | $-1$ | 1 | $-1$ | 1 | $-1$ | $-1$ | $-1$ |
| EMNIST | $-1$ | $-0.5$ | 0 | $-0.5$ | 0 | $-0.5$ | $-0.5$ | $-0.5$ |
| CIFAR-10 | $-3$ | $-2$ | 0 | $-2$ | 0 | $-2$ | $-1$ | $-2$ |
| CIFAR-100 | $-3$ | $-2$ | 0 | $-2$ | 0 | $-2$ | $-1$ | $-2$ |
| CINIC-100 | $-3$ | $-2$ | 0 | $-2$ | 0 | $-2$ | $-1$ | $-2$ |

Other hyperparameters are kept the same for all algorithms. In particular, we apply a weight decay of 0.0001 for all algorithms and decay $\eta_l$ by a factor of 0.998 in every round. We also use gradient clipping to improve stability of the algorithms as done in previous works (Acar et al., 2021). In all experiments we fix the number of participating clients to be 20, minibatch size to be 50 (for the synthetic dataset this reduces to full-batch gradient descent) and number of local updates $\tau$ to be 20.

## D.4 SENSITIVITY OF FEDEXP TO $\epsilon$

To evaluate the sensitivity of FedExP to $\epsilon$, we compute the training accuracy of FedExP after 500 rounds for varying $\epsilon$ and on different tasks. For each task, we fix $\eta_l$ to be the value used in our experiments in Section 6 and only vary $\epsilon$. The results are summarized below.

Table 4: Training accuracy obtained by FedExP with different choices of $\epsilon$ after 500 rounds of training on various tasks. Value of $\eta_l$ is fixed for each task ($10^{-0.5}$ for EMNIST and $10^{-2}$ for others). Results averaged over last 10 rounds.

| Dataset | $\epsilon = 10^{-3}$ | $\epsilon = 10^{-2.5}$ | $\epsilon = 10^{-2}$ | $\epsilon = 10^{-1.5}$ | $\epsilon = 10^{-1}$ |
|---|---|---|---|---|---|
| EMNIST | 85.40 | **86.26** | 85.73 | 85.49 | 84.90 |
| CIFAR-10 | **84.79** | 77.82 | 77.63 | 77.66 | 77.64 |
| CIFAR-100 | **59.01** | 44.76 | 44.21 | 44.37 | 44.40 |
| CINIC-10 | **66.31** | 60.93 | 61.05 | 60.47 | 60.96 |

We see that the sensitivity of $\epsilon$ is similar to that of the $\tau$ parameter which is added to the denominator of FedAdam and FedAdagrad (Reddi et al., 2021) to prevent the step size from blowing up.

Keeping $\epsilon$ too large reduces the adaptivity of the method and makes the behavior similar to `FedAvg`. At the same time, keeping $\epsilon$ too small may not also be beneficial always as seen in the case of EMNIST. In practice, we find that a grid search for $\epsilon$ in the range $\{10^{-3}, 10^{-2.5}, 10^{-2}, 10^{-1.5}, 10^{-1}\}$ usually suffices to yield a good value of $\epsilon$. A general rule of thumb would be to start with $\epsilon = 10^{-3}$ and increase $\epsilon$ till the performance drops.

### D.5 ADDITIONAL RESULTS

In this section, we provide additional results obtained from our experiments.

**Synthetic Linear Regression.** Note that for the synthetic linear regression experiments there is no test data. Also note that there is no randomness in this experiment since clients compute full-batch gradients with full participation. We provide the plot of $\eta_g^{(t)}$ for `FedExP` in Figure 7. We see that `FedExP` takes much larger steps in some (but not all) rounds compared to the constant optimum step size taken by our baselines, leading to a large speedup. Recall that we also let $\epsilon = 0$ in this experiment (since it aligns with our theory) which also explains the larger values of $\eta_g^{(t)}$ taken by `FedExP` in this case.

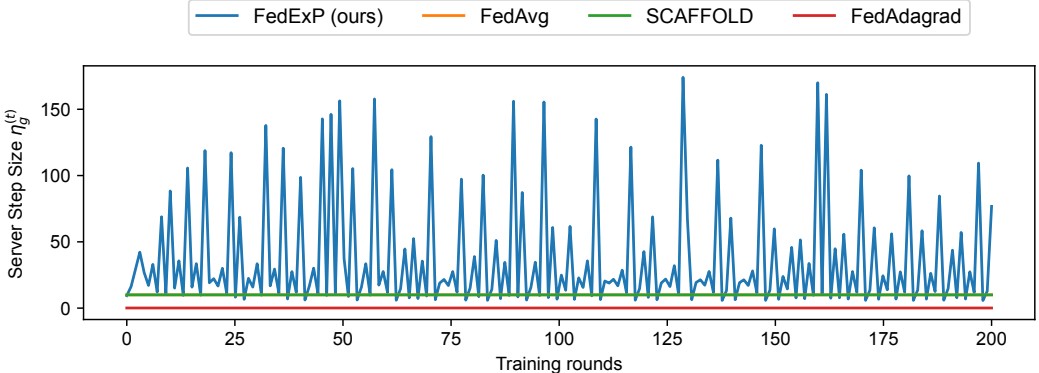

Figure 7: Global learning rates for synthetic data with linear regression. Results from a single instance of experiment.

**EMNIST.** For EMNIST we observe that `SCAFFOLD` gives slightly better training loss than `FedExP` towards the end of training. As described in Section 6, extrapolation can be combined with the variance-reduction in `SCAFFOLD` (the resulting algorithm is referred to as `SCAFFOLD-ExP`) to further improve performance. This gives the best result in this case as shown in Figure 8.

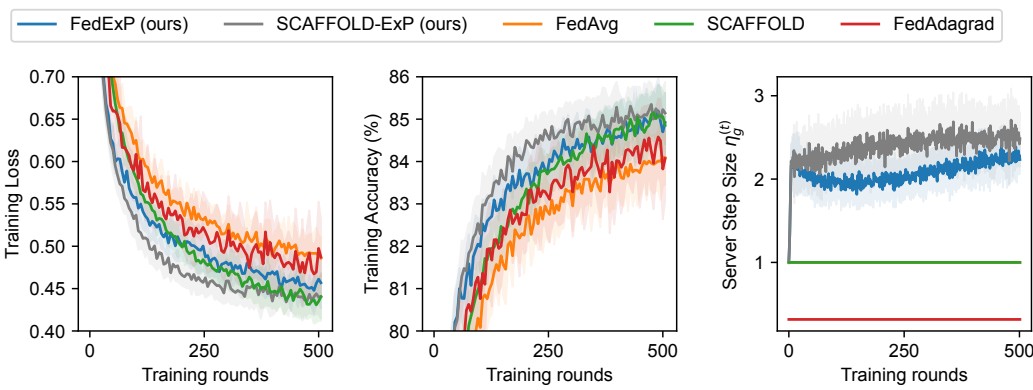

Figure 8: Additional results for EMNIST dataset. Mean and standard deviation from experiments with 20 different random seeds. The shaded areas show the standard deviation.

**CIFAR-10, CIFAR-100 and CINIC-10.** From Figure 3 and Figures 9–11, we see that `FedExP` comprehensively outperforms baselines in these cases, achieving almost $10\%$–$20\%$ higher accuracy than the closest baseline by the end of training. The margin of improvement is most in CIFAR-100, which can be considered as the toughest dataset in our experiments. This points to the practical utility of `FedExP` even in challenging FL scenarios.

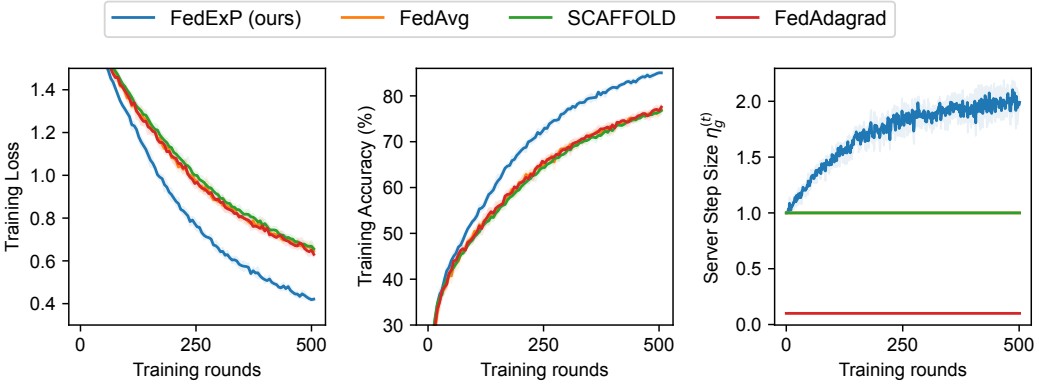

Figure 9: Additional results for CIFAR-10 dataset. Mean and standard deviation from experiments with 5 different random seeds. The shaded areas show the standard deviation.

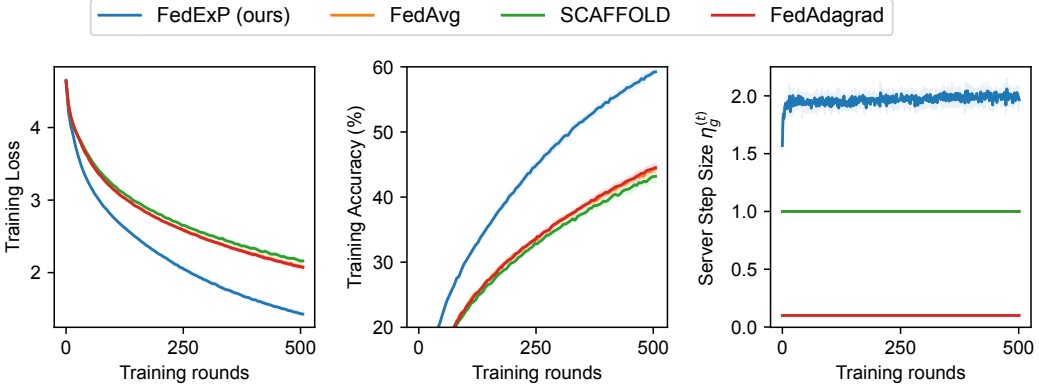

Figure 10: Additional results for CIFAR-100 dataset. Mean and standard deviation from experiments with 5 different random seeds. The shaded areas show the standard deviation.

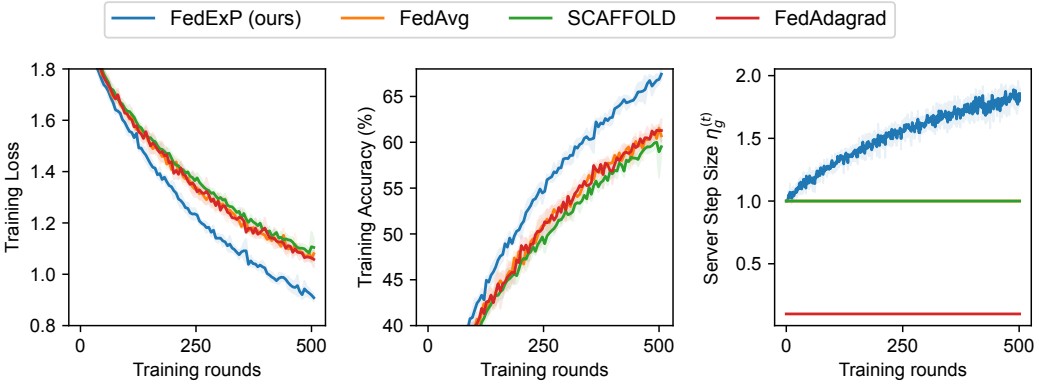

Figure 11: Additional results for CINIC-10 dataset. Mean and standard deviation from experiments with 5 different random seeds. The shaded areas show the standard deviation.

**Long-Term Behavior of Algorithms and Comparison with `FedProx`.** To evaluate the long-term behavior of different algorithms, we ran the experiments for 2000 rounds. Here, we also consider an additional algorithm, namely `FedProx`, for comparison. For fair comparison, we have tuned the $\mu$ parameter of `FedProx` for each dataset, by doing a grid search over the range $\{10^{-3}, 10^{-2}, 10^{-1}, 1\}$ as done in the original `FedProx` paper (Li et al., 2020). The results of EMNIST, CIFAR-10, CIFAR-100, and CINIC-10 in Figures 12–14 and Table 5 are from experiments with 3 different random seeds. Except for the synthetic dataset, the plots show mean and standard deviation values across all the random seeds and also over a moving average window of size 20.

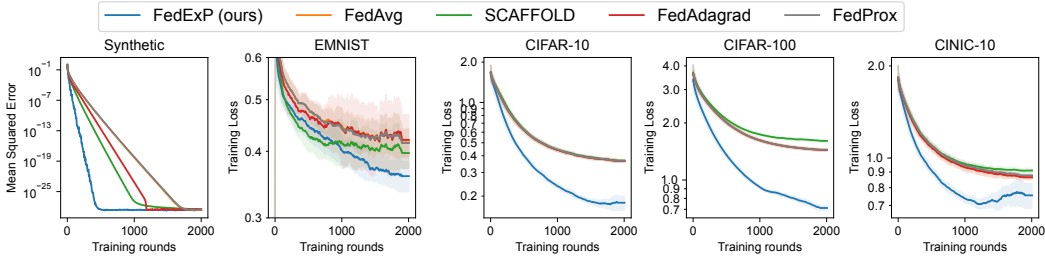

Figure 12: Training loss results of `FedExP`, `FedAvg`, `SCAFFOLD`, `FedAdagrad` and `FedProx` on the Synthetic, EMNIST, CIFAR-10,CIFAR-100 and CINIC-10 datasets for 2000 rounds.

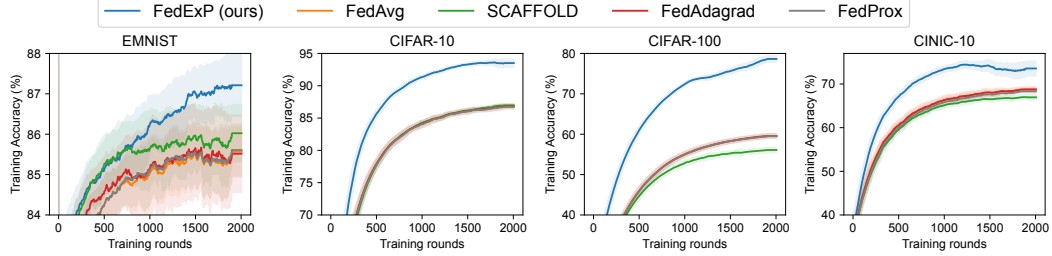

Figure 13: Training accuracy results of `FedExP`, `FedAvg`, `SCAFFOLD`, `FedAdagrad` and `FedProx` on the EMNIST, CIFAR-10, CIFAR-100 and CINIC-10 datasets for 2000 rounds.

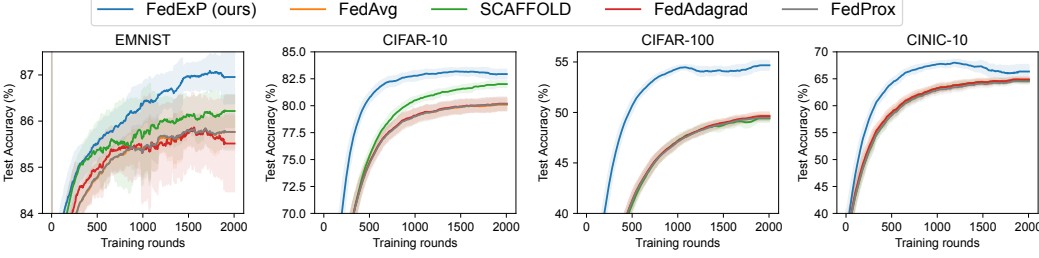

Figure 14: Test accuracy results of `FedExP`, `FedAvg`, `SCAFFOLD`, `FedAdagrad` and `FedProx` on the EMNIST, CIFAR-10, CIFAR-100 and CINIC-10 datasets for 2000 rounds.

Table 5: Test accuracy obtained by `FedExP` and baselines after 2000 rounds of training on various tasks. Results are averaged across 3 random seeds and last 20 rounds.

| Dataset | FedExP | FedAvg | SCAFFOLD | FedAdagrad | FedProx |
|---------|--------|--------|----------|------------|---------|
| EMNIST | $\mathbf{86.96} \pm 0.58$ | $85.78 \pm 0.35$ | $86.22 \pm 0.35$ | $85.53 \pm 1.04$ | $85.77 \pm 0.39$ |
| CIFAR-10 | $\mathbf{82.94} \pm 0.42$ | $80.10 \pm 0.56$ | $82.02 \pm 0.30$ | $80.21 \pm 0.60$ | $80.16 \pm 0.59$ |
| CIFAR-100 | $\mathbf{54.65} \pm 0.49$ | $49.63 \pm 0.37$ | $49.40 \pm 0.38$ | $49.64 \pm 0.39$ | $49.47 \pm 0.31$ |
| CINIC-10 | $\mathbf{66.45} \pm 1.28$ | $64.87 \pm 0.44$ | $64.61 \pm 0.49$ | $64.87 \pm 0.44$ | $64.52 \pm 0.45$ |

We see that `FedExP` continues to outperform baselines including `FedProx` in the long-term behavior as well.

# E COMBINING EXTRAPOLATION WITH SCAFFOLD

As described in Section 6, the extrapolation step can be added to the SCAFFOLD algorithm in a similar way as FedExP. The detailed steps of this SCAFFOLD-ExP algorithm are shown in Algorithm 2.

---

**Algorithm 2** SCAFFOLD-ExP

---

1: **Input:** $\mathbf{w}^{(0)}$, control variate $\mathbf{c}^{(0)}, \mathbf{c}_i^{(0)}, \forall i \in [M]$, number of rounds $T$, local iteration steps $\tau$, parameters $\eta_l, \epsilon$
2: **For** $t = 0, \ldots, T-1$ **communication rounds do**:
3:     **Global server do:**
4:     Send $\mathbf{w}^{(t)}, \mathbf{c}^{(t)}$ to all clients
5:     **Clients** $i \in [M]$ **in parallel do:**
6:       Set $\mathbf{w}_i^{(t,0)} \leftarrow \mathbf{w}^{(t,0)}$
7:       **For** $k = 0, \ldots, \tau-1$ **local iterations do:**
8:         Update $\mathbf{w}_i^{(t,k+1)} \leftarrow \mathbf{w}_i^{(t,k)} - \eta_l \left( \nabla F_i(\mathbf{w}_i^{(t,k)}, \xi_i^{(t,k)}) - \mathbf{c}_i^{(t)} + \mathbf{c}^{(t)} \right)$
9:       Compute $\Delta_i^{(t)} \leftarrow \mathbf{w}^{(t)} - \mathbf{w}_i^{(t,\tau)}$ and $\Psi_i^{(t)} \leftarrow \mathbf{c}^{(t)} - \frac{1}{\tau \eta_l} \Delta_i^{(t)}$
10:      Send $\Delta_i^{(t)}$ and $\Psi_i^{(t)}$ to the server
11:      Update local control variate $\mathbf{c}_i^{(t+1)} \leftarrow \mathbf{c}_i^{(t)} - \Psi_i^{(t)}$
12:     **Global server do:**
13:     Compute $\bar{\Delta}^{(t)} \leftarrow \frac{1}{M} \sum_{i=1}^{M} \Delta_i^{(t)}$ and $\eta_g^{(t)} \leftarrow \max \left\{ 1, \sum_{i=1}^{M} \left\| \Delta_i^{(t)} \right\|^2 \Big/ 2M \left( \left\| \bar{\Delta}^{(t)} \right\|^2 + \epsilon \right) \right\}$
14:     Update global model with $\mathbf{w}^{(t+1)} \leftarrow \mathbf{w}^{(t)} - \eta_g^{(t)} \bar{\Delta}^{(t)}$
15:     Compute $\bar{\Psi}^{(t)} \leftarrow \frac{1}{M} \sum_{i=1}^{M} \Psi_i^{(t)}$
16:     Update global control variate with $\mathbf{c}^{(t+1)} \leftarrow \mathbf{c}^{(t)} - \bar{\Psi}^{(t)}$

---

## F    COMBINING EXTRAPOLATION WITH SERVER MOMENTUM

We begin by recalling some notation from our work. The vector $\mathbf{w}^{(t)}$ is the global model at round $t$ and $\bar{\Delta}^{(t)}$ is the average of client updates at round $t$. The server momentum update at round $t$ can be written as $\mathbf{v}^{(t)} = \bar{\Delta}^{(t)} + \beta\mathbf{v}^{(t-1)}$ (let $\mathbf{v}^{-1} = \mathbf{0}$) and the global model update can be written as $\mathbf{w}^{(t+1)} = \mathbf{w}^{(t)} - \eta_g^{(t)}\mathbf{v}^{(t)}$. Our goal is now to find $\eta_g^{(t)}$ that minimizes $\left\|\mathbf{w}^{(t+1)} - \mathbf{w}^*\right\|^2$. We have,

$$\left\|\mathbf{w}^{(t+1)} - \mathbf{w}^*\right\|^2 = \left\|\mathbf{w}^{(t)} - \mathbf{w}^*\right\|^2 + (\eta_g^{(t)})^2\left\|\mathbf{v}^{(t)}\right\|^2 - 2\eta_g^{(t)}\langle\mathbf{w}^{(t)} - \mathbf{w}^*, \mathbf{v}^{(t)}\rangle. \tag{107}$$

Setting the derivative of the RHS of (107) to zero we have,

$$(\eta_g^{(t)})_{\text{opt}} = \frac{\langle\mathbf{w}^{(t)} - \mathbf{w}^*, \mathbf{v}^{(t)}\rangle}{\left\|\mathbf{v}^{(t)}\right\|^2}. \tag{108}$$

Our goal now is to find a lower bound on $\langle\mathbf{w}^{(t)} - \mathbf{w}^*, \mathbf{v}^{(t)}\rangle$. We have the following lemma.

**Lemma 11.** *Assume that* $\left\langle\mathbf{w}^{(t)} - \mathbf{w}^*, \bar{\Delta}^{(t)}\right\rangle \geq m^{(t)} = \sum_{i=1}^M\left\|\Delta_i^{(t)}\right\|^2/M$ *(see Appendix C.4.1) for all* $t \geq 0$ *and* $\eta_g^{(r)} \leq (m^{(r)} + \sum_{k=0}^{r-1}(\beta/2)^{r-k}m^{(k)})/2\left\|\mathbf{v}^{(r)}\right\|^2$ *for all* $r < t - 1$. *Then,*

$$\left\langle\mathbf{w}^{(t)} - \mathbf{w}^*, \mathbf{v}^{(t)}\right\rangle \geq m^{(t)} + \sum_{k=0}^{t-1}(\beta/2)^{t-k}m^{(k)}, \tag{109}$$

*which implies,*

$$(\eta_g^{(t)})_{opt} \geq \frac{m^{(t)} + \sum_{k=0}^{t-1}(\beta/2)^{t-k}m^{(k)}}{2\left\|\mathbf{v}^{(t)}\right\|^2}. \tag{110}$$

**Proof.** We proceed via a proof by induction. The statement clearly holds at $t = 0$ since $\left\langle\mathbf{w}^{(0)} - \mathbf{w}^*, \mathbf{v}^{(0)}\right\rangle = \left\langle\mathbf{w}^{(0)} - \mathbf{w}^*, \bar{\Delta}^{(0)}\right\rangle \geq m^{(0)}$.

Now assuming the lemma holds at $t - 1$ we have,

$$\left\langle\mathbf{w}^{(t)} - \mathbf{w}^*, \mathbf{v}^{(t)}\right\rangle = \left\langle\mathbf{w}^{(t)} - \mathbf{w}^*, \bar{\Delta}^{(t)}\right\rangle + \beta\left\langle\mathbf{w}^{(t)} - \mathbf{w}^*, \mathbf{v}^{(t-1)}\right\rangle \tag{111}$$

$$= \left\langle\mathbf{w}^{(t)} - \mathbf{w}^*, \bar{\Delta}^{(t)}\right\rangle + \beta\left\langle\mathbf{w}^{(t-1)} - \eta_g^{(t-1)}\mathbf{v}^{(t-1)} - \mathbf{w}^*, \mathbf{v}^{(t-1)}\right\rangle \tag{112}$$

$$\geq m^{(t)} + \beta\left[\left\langle\mathbf{w}^{(t-1)} - \mathbf{w}^*, \mathbf{v}^{(t-1)}\right\rangle - \eta_g^{(t-1)}\left\|\mathbf{v}^{(t-1)}\right\|^2\right] \tag{113}$$

$$\geq m^{(t)} + \sum_{k=0}^{t-1}(\beta/2)^{t-k}m^{(k)}, \tag{114}$$

where the last line follows from the fact that $\left\langle\mathbf{w}^{(t-1)} - \mathbf{w}^*, \mathbf{v}^{(t-1)}\right\rangle \geq m^{(t-1)} + \sum_{k=0}^{t-2}(\beta/2)^{t-1-k}m^{(k)}$ and $\eta_g^{(t-1)} \leq (m^{(t-1)} + \sum_{k=0}^{t-2}(\beta/2)^{t-1-k}m^{(k)})/2\left\|\mathbf{v}^{(t-1)}\right\|^2$. $\qquad\square$

Thus we propose to keep the following server step size when using server momentum,

$$\eta_g^{(t)} = \frac{m^{(t)} + \sum_{k=0}^{t-1}(\beta/2)^{t-k}m^{(k)}}{2(\left\|\mathbf{v}^{(t)}\right\|^2 + \epsilon)}, \tag{115}$$

where $m^{(t)} = \sum_{i=1}^M\left\|\Delta_i^{(t)}\right\|^2/M$. Note that we also add a small constant $\epsilon$ to the denominator to prevent the step size from blowing up as done for FedExP. We call server momentum with this step size as FedExP-M.

We compare the performance of FedExP-M with FedAdam and FedAvg-M (FedAvg with server momentum) on the CIFAR-10 and CIFAR-100 datasets as shown in Figures 15–17, where the mean and standard deviation values are computed over 3 random seeds and a moving average window of

size 20. The experimental setup is the same as described in Section 6. The hyperparameters $\eta_l, \epsilon$ for `FedExP-M` and $\eta_l, \eta_g$ for `FedAdam` and `FedAvg-M` were tuned following a similar process as described in Appendix D.3, and their resulting values are in Table 6.

Table 6: Base-10 logarithm of the best combination of $\epsilon$ and $\eta_l$ for `FedExP-M` and combination of $\eta_l$ and $\eta_g$ for `FedAdam` and `FedAvg-M`.

| Dataset | FedExP | | FedAdam | | FedAvgm-M | |
|---|---|---|---|---|---|---|
| | $\epsilon$ | $\eta_l$ | $\eta_g$ | $\eta_l$ | $\eta_g$ | $\eta_l$ |
| CIFAR-10 | $-3$ | $-2$ | $-2$ | $-2$ | $0$ | $-2$ |
| CIFAR-100 | $-3$ | $-2$ | $-2$ | $-2$ | $0$ | $-2$ |

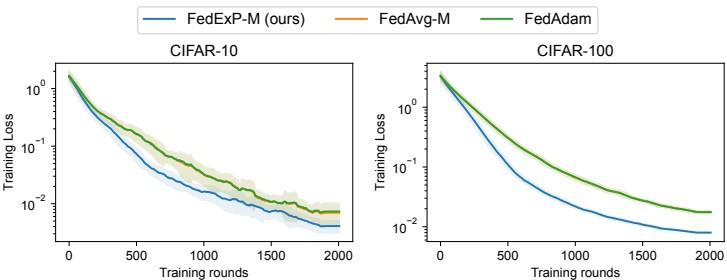

Figure 15: Training loss results of `FedExP-M`, `FedAdam` and `FedAvg-M` on the CIFAR10 and CIFAR100 datasets.

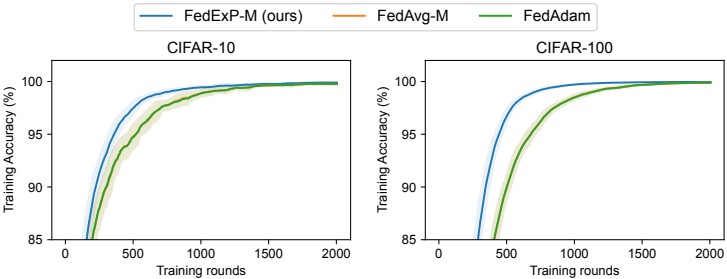

Figure 16: Training accuracy results of `FedExP-M`, `FedAdam` and `FedAvg-M` on the CIFAR10 and CIFAR100 datasets.

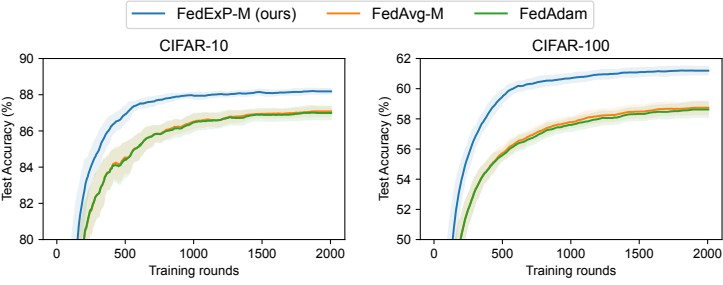

Figure 17: Test accuracy results of `FedExP-M`, `FedAdam` and `FedAvg-M` on the CIFAR10 and CIFAR100 datasets.

Our result shows that server momentum can be successfully combined with extrapolation for the best speed-up among all baselines. The behavior of `FedAdam` and `FedAvg-M` are quite similar in these experiments which can be attributed to the dense nature of the gradients in image classification as

discussed in Section 6. We note that this is only a preliminary result and future work will look to study the effect of combining server momentum and extrapolation more rigorously.

