# OpenReview forum: "FedExP: Speeding Up Federated Averaging via Extrapolation"
_ICLR.cc/2023/Conference — ICLR 2023 notable top 25%_

### Official Review · Reviewer_Qfnz · 2022-10-25

**Confidence:** 3
**Correctness:** 4
**Technical Novelty And Significance:** 3
**Empirical Novelty And Significance:** 3
**Recommendation:** 8

**Clarity, Quality, Novelty And Reproducibility:**

- The paper is well-written and easy to follow. Contributions and limitations are made quite clear. The work quality is good and it is fairly original.

- Experimental result for EMNIST looks contradicting to the Adaptive FedOpt work (Reddi et al., 2021) as in the original paper SCAFFOLD performed significantly worse than FedAvg and FedAdagrad. Could you please comment on what could be the reasons for these different behaviors?

- I would like to as the authors why the proposed method was not compared to FedAdam which achieves the best empirical performance on some of the benchmarks. I understand that it does not have such a nice theory, but still think that it would be very useful for practitioners to include it in the comparison at least in the appendix (or how it can be combined with FedExP).

- How well can FedExP be combined with client and server momentum techniques?

- How sensitive is the proposed method to the choice of parameter $\epsilon$? Does it require a lot of tuning for achieving good performance or the same value can be used across a variety of settings like for some Adam parameters? I think that an experiment or at least a comment on this is needed for publication.

**Minor comments and typos**

- I do not quite get the phrase from the Abstract
> practical benefit of the server step size has not been seen in most existing works

The works of Hsu et al., 2019 and Reddi et al., 2021 address this question, don't they?

- Why the number of local updates is fixed to 20 instead of 1 epoch over the local data like it was done in the paper of Reddi et al., 2021?

- It would be helpful to include the result of Khaled et al., 2020 on local GD (FedAvg) in the main body for a clearer comparison to FedExP. I would also like to ask the authors to comment and point to (also add it to the Appendix) the adjustments they made to the analysis from previous works.

- Probably missed word on page 7 (Figure 2 caption): *"the last iterate of has an oscillating"*.

- Looks like there is an issue in this sentence *"FedExP is now trying to minimize $\Delta_2^{(t+1)}$"*, as it is not clear how a vector can be minimized.

**Strength And Weaknesses:**

## Strengths

1. Novel and insightful view of an important Federated Averaging algorithm, which has the potential for fruitful theoretical and practical advancements in the area of federated optimization.

3. The suggested step-size strategy can be applicable to a wider set of local gradient methods, which was illustrated for SCAFFOLD in the Appendix.

2. Thorough empirical study of the proposed method, which shows its insightful properties on toy problems and promising results for synthetic and realistic benchmarks.

## Weaknesses

1. Convergence analysis focuses on the local full-batch and full client participation. which is quite far from current practical federated learning settings. Although I do not consider it a big issue, as almost all local optimization methods do not provide any theoretical benefits over simplest distributed (mini-batch stochastic) gradient descent baseline for heterogenous problems.

**Summary Of The Paper:**

This paper studies the problem of federated optimization. It shows a connection between FedAvg and Projection Onto Convex Sets algorithm (under overparametrized convex setting) and based on this proposes a method (FedExP) with adaptive server step-size strategy by using the gradient diversity measure. The authors perform a theoretical analysis of FedExP and show convergence under standard reasonable assumptions in convex and non-convex settings. In the experimental section proposed method is studied in both illustrative and realistic FL settings and is shown to outperform selected baselines.

**Summary Of The Review:**

This work is insightful from theoretical perspective and presents useful practical results. I have some comments and questions (in the previous section) on parts of the paper, which will be hopefully addressed. Overall, I think that submission is worth publishing after the minor revision.

---

> ### Author Response · Authors · 2022-11-16
> **Response to Reviewer Qfnz Continued**
>
>
> * **Sensitivity of FedExP to $\epsilon$:** Great question. To evaluate the sensitivity of FedExP to $\epsilon$, we compute the training accuracy of FedExP after 500 rounds for varying $\epsilon$ and on different tasks. For each task, we fix $\eta_l$ to be the value used in our experiments in Section 6 and only vary $\epsilon$. The results are summarized below. The results are also added to Appendix D.4 in our revised version.
>
>   [Table 2: Training accuracy obtained by FedExP with different choices of $\epsilon$ after 500 rounds of training on various tasks. Value of $\eta_l$ is fixed for each task ($10^{-0.5}$ for EMNIST and $10^{-2}$ for others). Results averaged over the last 10 rounds.]
>     | Dataset | $\epsilon=10^{-3}$ | $\epsilon=10^{-2.5}$ | $\epsilon = 10^{-2}$ | $\epsilon = 10^{-1.5}$ | $\epsilon=10^{-1}$|
>     | :---    |:----:          | :----:          |  :----: | :----:    | ---:   |
>     | EMNIST  |$85.40$         | $\mathbf{86.26}$| $85.73$ | $85.49$   | $84.90$|
>     | CIFAR-10|$\mathbf{84.79}$| $77.82$|  $77.63$| $77.66$| $77.64$|
>     |CIFAR-100|$\mathbf{59.01}$| $44.76$| $44.21$| $44.37$| $44.40$|
>     | CINIC-10| $\mathbf{66.31}$| $60.93$|  $61.05$| $60.47$| $60.96$|
>
>   We see that the sensitivity of $\epsilon$ is similar to that of the $\tau$ parameter that is added to the denominator of FedAdam and FedAdagrad [1] to prevent the step size from blowing up (see results on page 36). Keeping $\epsilon$ too large reduces the adaptivity of the method and makes the behavior similar to FedAvg. At the same time, keeping $\epsilon$ too small may not also be beneficial always as seen in the case of EMNIST.  In practice, we find that a grid search for $\epsilon$ in the range $[10^{-3},10^{-2.5},10^{-2},10^{-1.5},10^{-1}]$ usually suffices to yield a good value of $\epsilon$. A general rule of thumb would be to start with $\epsilon = 10^{-3}$ and increase $\epsilon$ till the performance drops. Thus, while there does not exist a universally good value of $\epsilon$, we believe it does not require a lot of tuning.
>
>
>
>
>
>
> ### **Re: Minor Points**
>
> * **Practical benefit of server step size:** While previous works have introduced the concept of a server step size, we believe they have not been able to show a consistent practical benefit over vanilla averaging. For instance, Reddi et al. perform extensive hyperparameter tuning for FedAvg on 6 different datasets and find that just vanilla averaging ($\eta_g=1$)  with properly tuned $\eta_l$ performs the best in 5 out of 6 experiments (see best performing hyperparameters on page 29 in [1]).  Our goal was to convey this observation through the line you cited in the abstract. We are happy to rephrase this line if you think it needs more explanation.
>
>
> ***
>
> * **Number of local steps fixed to be 20:** In the case where clients have different dataset sizes (e.g. EMNIST), performing a fixed number of epochs with a fixed batch size can lead to each client performing a different number of local SGD steps. As shown in [2], this leads to an objective inconsistency problem and requires an additional correction to fix. Therefore for simplicity, we fix the number of local steps as 20 for all clients.
>
> ***
>
> * **Including the result of Khaled et al. and pointing out adjustments:** Thank you for the suggestion. We have modified the discussion in Section 3 to explicitly state the convergence result of FedAvg obtained by previous works. We have also included additional discussion in Appendix C.2. and Appendix C.3. on the adjustments we made to our analysis from previous works.
>
> ***
>
> * **Typo in Fig. 2 caption and issue in the sentence on page 7:** We have fixed both of these. Thank you for the careful proofreading!
>
>
> ***
>
>
> We hope that our response addresses your concerns. We are happy to answer any further questions that you may have. Thank you!
>
>
>
> **References**
>
> [1] Reddi, S., Charles, Z., Zaheer, M., Garrett, Z., Rush, K., Konečný, J., ... & McMahan, H. B. (2020). Adaptive federated optimization. arXiv preprint arXiv:2003.00295.
>
> [2] Wang, J., Liu, Q., Liang, H., Joshi, G., & Poor, H. V. (2020). Tackling the objective inconsistency problem in heterogeneous federated optimization. Advances in neural information processing systems, 33, 7611-7623.

---

> ### Author Response · Authors · 2022-11-16
> **Response to Reviewer Qfnz**
>
> We greatly appreciate the reviewer’s positive comments on the insightfulness of our proposed approach, applicability to a wider set of local gradient methods and thorough empirical study. We address the weaknesses mentioned and clarify the questions asked below. We have also revised our paper based on your suggestions and highlighted all major changes in blue.
>
>
>
>
>
> ### **Re: Weaknesses**
>
> * **Convergence analysis focuses on local full batch and full client participation:** We acknowledge that this is currently a theoretical limitation of our work. As discussed in the paper, this is due to the difficulty in decoupling the effect of stochastic noise on the server step size and client pseudo-gradients. In practice, FedExP works well even with stochastic noise as seen in our experiments. Future work will look to strengthen our convergence analysis to include stochastic noise as well.
>
>
>
> ### **Re: Clarity, Quality, Novelty And Reproducibility**
>
> * **Experiment results for EMNIST look contradicting to Reddi et al.:** Thank you for pointing this out. A closer inspection reveals that there are two key differences in our experimental setting versus that of Reddi et al. -  i) number of local steps (20 steps vs 1 epoch), ii) number of participating clients (20 vs 10). To the best of our knowledge, Reddi et al. have not provided the hyperparameters they use for SCAFFOLD and their open-source code does not contain an implementation of SCAFFOLD, so it is not possible for us to exactly replicate their result to understand the difference. We conjecture that the biggest reason for the difference in results is the smaller number of clients sampled per round in Reddi et al., which increases the staleness of the control variates in SCAFFOLD.
>
> ***
>
> * **Comparison with FedAdam and combining with server momentum:** Indeed, FedAdam achieves the best empirical results on some benchmarks by combining Adagrad-style adaptive server step sizes with server momentum. We believe the improvement is largely due to server momentum, since just varying the step size (as done in Adagrad) does not yield significant improvement, as seen in our experiments. Server momentum is an orthogonal technique to extrapolation and can be combined with extrapolation as we discuss below.
>
>   We begin by recalling some notation from our work. $\mathbf{w}^{(t)}$ is the global model at round $t$ and $\bar{\Delta}^{(t)}$ is the average of client updates at round $t$. The server momentum update at round $t$ can then be written as $\mathbf{v}^{(t)} = \bar{\Delta}^{(t)} + \beta \mathbf{v}^{(t-1)}$ and the global model update can be written as $\mathbf{w}^{(t+1)} = \mathbf{w}^{(t)}-\eta_g^{(t)}\mathbf{v}^{(t)}$. We follow a similar procedure as done for FedAvg, to propose the following lower bound on the value of $(\eta_g^{(t)})^*$ that minimizes $\||\mathbf{w}^{(t+1)} - \mathbf{w}^*\||^2$:
>
>   $$(\eta_g^{(t)})^* \geq \frac{m^{(t)} + \sum_{k=0}^{t-1}(\beta/2)^{t-k}m^{(k)}}{2(\||\mathbf{v}^{(t)}\||^2 + \epsilon)}$$
>
>   where $m^{(k)} = \sum_{i=1}^M \||\Delta_i^{(t)}\||^2/M$. Details of the exact derivation can be found in Appendix E. We term server momentum with this step-size as FedExP-M. We compare FedExP-M with FedAdam and FedAvg-M (FedAvg with server momentum) on the CIFAR-10 and CIFAR-100 datasets as shown below. The experimental setup is the same as described in Section 6. Hyperparameters $\eta_l,\epsilon$ for FedExP-M and $\eta_l,\eta_g$ for FedAdam and FedAvg-M are tuned following a similar process as described in Appendix D.3.
>
>   [Table 1: Test accuracy obtained by FedExP and baselines after 2000 rounds of training on the EMNIST, CIFAR-10, CIFAR-100 and CINIC-10 datasets. Results averaged over 3 random seeds.]
>     | Dataset    | FedExP-M                           | FedAdam             | FedAvg-M             |
>     | :---           | :----:                                     | :----:                     | ---:                        |
>     | CIFAR-10 | $\mathbf{88.18} \pm 0.14$ | $87.00 \pm 0.36$|  $87.07 \pm 0.26$|
>     |CIFAR-100| $\mathbf{61.18} \pm 0.26$ | $58.63 \pm 0.49$| $58.73 \pm 0.42$ |
>
>
>
>   We have also included the training,test accuracy plots and best performing hyperparameters in Appendix E. Our result shows that server momentum can be successfully combined with extrapolation for the best speed-up among all baselines. We note that this is only a preliminary result and future work will look to study the effect of combining server momentum and extrapolation more rigorously.
> ***
>
> * **Combining with client momentum:** We do not foresee any difficulty in combining extrapolation with client momentum as client momentum only affects the local optimization at clients and does not affect the server aggregation step.

---

### Official Review · Reviewer_AfRg · 2022-10-25

**Confidence:** 4
**Correctness:** 4
**Technical Novelty And Significance:** 3
**Empirical Novelty And Significance:** 3
**Recommendation:** 8

**Clarity, Quality, Novelty And Reproducibility:**

I found the paper to be written very clearly. Some minor issues/typos:
Algorithm  1: "Global server do" -> "Global server does" or maybe "On global server do"?
equations (1), (8), (10) and (11) should end with a comma
Page 7, "has a oscillating" -> "has an oscillating"
Equations (71) and (72) include $\nabla \mathbf{h}_i^{(t)}$, which seems to be a typo, it should have been $\mathbf{h}_i^{(t)}$
What is the point of introducing $\mathbf{h}_i^{(t)}$ in the proof of Theorem 2, isn't it the same as $\Delta_i^{(t)}$?
What's the point of dividing by $\eta_g^{(t)}$ in (73) if you then multiply back by $\eta_g^{(t)}$ to get (76)?

**Strength And Weaknesses:**

## Strength
1. The work is written in a clear and intuitive way.
2. The considered problem is quite meaningful as server side stepsizes have been shown to give an improvement in practice and building better adaptive estimates is a promising direction.
3. The theory is rigorous and considers both convex and nonconvex settings.
4. There are experiments to support the theory as well as to show that the method will be useful in practice.

## Weaknesses:
1. In the experiments, it appears that most algorithms were terminated before convergence. Therefore, we do not see the final values of the test accuracy. In practice, these values are very important, so I would like to know if the authors can present plots with longer training.
2. Theorem 2 is analyzed under the assumption that the gradients are deterministic and I do not see if this stated anywhere in the text.


**Summary Of The Paper:**

This paper studies adaptive server step sizes for federated learning (FL). The authors propose a method called FedExP, which uses a server step size akin to the POCS algorithm from the literature on finding feasible points inside the intersections of sets. The authors use a similar expression to estimate how much one can extrapolate in the direction proposed by FedAvg.

The authors prove convergence of the method for convex and nonconvex problems. They also run experiments on convex problems that clearly illustrate the theory, as well as on FL benchmarks.

-------------
## update after reading authors' feedback

The authors addressed my main concern about the short runs. I can see the figures in the appendix that include longer runs.
The authors also added the missing assumptions to Theorem 2.
I believe it is very clear that the paper should be accepted.

**Summary Of The Review:**

The paper shows a good mixture of theoretical and empirical results, both of which seem of sufficient quality. There are a couple of ways the paper can be improved, but I believe that it meets the bar for acceptance.

---

> ### Author Response · Authors · 2022-11-16
> **Response to Reviewer AfRg**
>
> We greatly appreciate the reviewer’s positive comments on the clarity of our writing, meaningfulness of the problem considered, rigorous theory and experimentation. We address the weaknesses and minor comments below. We have also revised our paper based on your suggestions and highlighted all major changes in blue.
>
>
>
> ### **Re: Weaknesses**
>
> * **Algorithms terminated before convergence:** Thank you for pointing this out. We provide plots in Appendix D.4 (page 30) with training up to 2000 rounds, by which time we observe that most algorithms have converged. We retain plots showing the short-term behavior (500 rounds) of algorithms in the main paper as it better visualizes the performance of different algorithms in the initial rounds. We also compare with an additional baseline, FedProx [1] as suggested by Reviewer LWDX. We summarize the test accuracies achieved by all the algorithms after 2000 rounds in the table below.
>
>    [Table 1: Test accuracy obtained by FedExP and baselines after 2000 rounds of training on the EMNIST, CIFAR-10, CIFAR-100 and CINIC-10 datasets. Results averaged over 3 random seeds and last 20 rounds.]
>
>     | Dataset | FedExP | FedAvg | SCAFFOLD| FedAdagrad| FedProx|
>     | :---    | :----: | :----: |  :----: | :----:    | ---:   |
>     | EMNIST  | $\mathbf{87.26} \pm 0.12$    | $85.98 \pm 0.10$|  $86.42 \pm 0.12$|    $85.94 \pm 0.62$| $86.03 \pm 0.08$|
>     | CIFAR-10| $\mathbf{82.99} \pm 0.20$ | $80.12 \pm 0.45$|  $82.14 \pm 0.37$|    $80.28 \pm 0.61$| $80.11 \pm 0.63$|
>     |CIFAR-100| $\mathbf{54.61} \pm 0.60$    | $49.61 \pm 0.26$|  $49.36 \pm 0.51$|    $49.59 \pm 0.27$| $49.59 \pm 0.18$|
>     | CINIC-10| $\mathbf{67.91} \pm 0.72$    | $65.15 \pm 0.22$|  $64.54 \pm 0.31$|    $65.14 \pm 0.18$| $65.76 \pm 0.11$|
>
>   We see that FedExP continues to outperform baselines including FedProx in   long-term behavior as well. We hope this addresses your concern.
>
>
>
> ***
>
> * **Theorem 2 assumption missing from the text:** We have modified the theorem statement to clearly state that clients compute full batch gradients and always participate in our convergence proof.
>
> ### **Re: Minor Issues/Typos**
>
> Thank you for the careful proofreading. We have incorporated your suggestion for Algorithm 1, put commas after equations (1), (8), (10), (11), fixed the typos in Page 7, Equation (71) and (72) and removed the redundancy in Equation (73).
>
> In the proof of Theorem 2, $\mathbf{h}_i^{(t)}$ is the *normalized* gradient and is defined as $\mathbf{h}_i^{(t)} = \Delta_i^{(t)}/\tau$. We introduce $\mathbf{h}_i^{(t)}$ just for convenience in our proof.
>
> ***
>
> We hope that our response addresses your concerns. We are happy to answer any further questions that you may have. Thank you!
>
> **References**
>
> [1] Li, T., Sahu, A. K., Zaheer, M., Sanjabi, M., Talwalkar, A., & Smith, V. (2020). Federated optimization in heterogeneous networks. Proceedings of Machine Learning and Systems, 2, 429-450.

---

> > ### Comment · Reviewer_AfRg · 2022-11-19
> > **Great!**
> >
> > I thank the authors for incorporating my feedback into the updated version of the paper. I'm very positive about the paper and I'm glad to see that all reviewers unanimously voted that it should be accepted.

---

> > > ### Author Response · Authors · 2022-11-21
> > > **Thank you!**
> > >
> > > Thank you once again for your insightful suggestions. We are grateful to have your strong support for our paper.

---

### Official Review · Reviewer_LWDX · 2022-10-27

**Confidence:** 4
**Correctness:** 4
**Technical Novelty And Significance:** 3
**Empirical Novelty And Significance:** 3
**Recommendation:** 8

**Clarity, Quality, Novelty And Reproducibility:**

Quality:
- Recently proposed federated learning algorithm FedProx has tackled the data heterogeneity problem by adapting the optimization objective of FedAvg on local clients. Although FedProx increases computation on client-side, it still has similar motivation. So, it would be interesting to see the comparison of FedExp with FedProx method.

Clarity:
- A figure of the proposed FedExP model with client-server communication would improve understanding of the method setup and working.
- A table of notations would improve readability of the mathematical formulation.

Novelty:
- Drawing a connection between FedAvg method and POCS algorithm is novel. Furthermore, adaptation and application of EPPM extrapolation algorithm in FedAvg method for adaptive server step size in each round is novel too.
- novelty is simple yet effective application in federated learning paradigm

Reproducibility:
- Experimental setup and hyperparameter settings are described in detail.


**Strength And Weaknesses:**

Strengths:
- novelty is simple yet effective application in federated learning paradigm
- The motivation is well founded and the claims are sound.
- Paper is clearly presented and easy to follow.
- Mathematical formulation is very detailed and explanatory.
- Proposed FedExP model is effective and efficient.
- Proposed FedExP model consistently outperforms two strong federated learning baseline methods on global model convergence time and test accuracy over multiple datasets.

Weaknesses:
- inspired by related works, especially the application of extrapolation in federated learning
- novelty is limited (yet effective)
- Missing comparison with the relevant FedProx method.
	Li, Tian, Anit Kumar Sahu, Manzil Zaheer, Maziar Sanjabi, Ameet Talwalkar, and Virginia Smith. "Federated optimization in heterogeneous networks." Proceedings of Machine Learning and Systems 2 (2020): 429-450.



**Summary Of The Paper:**

- In federated learning settings, aggregating the learning of all participating clients has always been a challenging issue in cases of data heterogeneity across clients which is very probable in real world scenarios. To tackle this, recent works have optimized the server aggregation process of FedAvg algorithm by treating client updates as "pseudo gradients" with fixed server step size. However, using a fixed step size is not ideal for varying data distribution across clients.
- To tackle this issue, this paper proposes a novel federated learning algorithm FedExP which adaptively determines the server step size in each round of federated learning such that it can handle varying degree of similarity in data distribution across clients. In doing so, the authors first establish a novel connection between FedAvg and POCS algorithms for overparameterized convex objectives and further draw the inspiration from the EPPM extrapolation algorithm (used to speed up POCS algorithm) resulting in adaptively determining server step size in FedExP for better and fast convergence of global model.
- Key feature of FedExP is that there is virtually no requirement for additional communication, computation, or storage at clients or the server.
- Experimental evaluation demonstrates that FedExP algorithm converges faster and consistently outperforms baselines over 5 datasets.
- novelty is simple and limited yet effectively applied to federated learning paradigm and have shown improved performance in terms of speed-up/efficiency and client-server data communication



**Summary Of The Review:**

The dynamic tuning of server step size results in efficient (faster) convergence and performance boost across multiple datasets while efficiently tackling the issue of knowledge interference during server aggregation due to heterogeneous data distribution across clients.

- The novelty is limited and simple yet effective in Federated learning paradigm
- the proposed method is effective both in terms on efficiency and prediction performance
- addresses the sustainability concerns and may be applicable and scale when a number of clients participate in Federated averaging, especially in IOT scenarios and cloud-edge continuum

---

> ### Author Response · Authors · 2022-11-16
> **Response to Reviewer LWDX**
>
> We greatly appreciate the reviewer’s positive comments on the simplicity and effectiveness of our proposed approach, clarity of presentation, and strong empirical performance. We address the weaknesses mentioned and suggestions for clarity below. We have also revised our paper based on your suggestions and highlighted all major changes in blue.
>
>
>
> ### **Re: Weaknesses**
>
> * **Inspired by related works and limited novelty:** We agree that our algorithm is inspired by the EPMM algorithm used to speed up POCS.  At the same time, we would like to emphasize that EPMM was originally designed for a fundamentally different problem and our key contribution is to identify the connection between FedAvg and POCS, which has been appreciated by all reviewers including yourself. We believe this connection has the potential to further inspire practical and theoretical advancements in FL, as also noted by Reviewer Qfnz. We also provide novel convergence guarantees for FedExP and provide new insight into how iterate averaging can benefit extrapolation via a toy example.
>
> ***
>
> * **Comparison with FedProx[1]:** We are happy to provide a comparison with FedProx. FedProx falls into the class of algorithms that aim to speed up FL by modifying the local optimization at clients to tackle the client-drift error. While theoretically attractive, we find that just reducing client drift might be insufficient to speed up FL in practice, as seen by the results of SCAFFOLD. We find that the results of FedProx also follow a similar trend. We have added plots showing the training and test accuracy of FedProx compared to FedExP and our original baselines in Appendix D.4. For fair comparison, we have tuned the $\mu$ parameter of FedProx for each dataset, by doing a grid search over the range $[10^{-3}, 10^{-2},10^{-1},1]$ as done in the original FedProx paper [1]. We summarize the test accuracies achieved by all the algorithms after 2000 rounds in the table below.
>
>   [Table 1: Test accuracies obtained by FedExP and baselines after 2000 rounds of training on the EMNIST, CIFAR-10, CIFAR-100, and CINIC-10 datasets. Results averaged over 3 random seeds and the last 20 rounds.]
>
>     | Dataset | FedExP | FedAvg | SCAFFOLD| FedAdagrad| FedProx|
>     | :---    | :----: | :----: |  :----: | :----:    | ---:   |
>     | EMNIST  | $\mathbf{86.96} \pm 0.58$    | $85.78 \pm 0.34$|  $86.22 \pm 0.35$|    $85.53 \pm 1.04$| $85.77 \pm 0.39$|
>     | CIFAR-10| $\mathbf{82.94} \pm 0.42$ | $80.10 \pm 0.56$|  $82.02 \pm 0.30$|    $80.20 \pm 0.59$| $80.16 \pm 0.59$|
>     |CIFAR-100| $\mathbf{54.65} \pm 0.49$    | $49.63 \pm 0.37$|  $49.40 \pm 0.38$|    $49.64 \pm 0.39$| $49.47 \pm 0.31$|
>     | CINIC-10| $\mathbf{66.45} \pm 1.28$    | $64.87 \pm 0.44$|  $64.61 \pm 0.49$|    $64.87 \pm 0.44$| $65.52 \pm 0.45$|
>
>
>
>   We would like to note that it is also possible to combine extrapolation with FedProx for faster convergence, as was done for SCAFFOLD for (see Section 6 and Appendix D).
>
> ***
>
> ### **Re: Clarity suggestions**
>
> Thank you for the suggestions on improving the clarity of our paper. We have included a table of notation and a schematic of the client-server communication in FedExP in Appendix B and mentioned this in the main paper at the end of Section 2.
>
> ***
>
> We hope that our response addresses your concerns. We are happy to answer any further questions that you may have. Thank you!
>
> **References**
>
> [1] Li, T., Sahu, A. K., Zaheer, M., Sanjabi, M., Talwalkar, A., & Smith, V. (2020). Federated optimization in heterogeneous networks. Proceedings of Machine Learning and Systems, 2, 429-450.

---

### Decision · Program_Chairs · 2023-01-20

**Decision:**

Accept: notable-top-25%

**Justification For Why Not Higher Score:**

The novelty is to some extent limited. The algorithm is simple and the analysis is based on standard technique and assumptions.

**Justification For Why Not Lower Score:**

The algorithm is effectiveness and the theoretical analysis, although it is quite standard, is convincing to prove the convergence.

**Metareview: Summary, Strengths And Weaknesses:**

The authors developed FedExp, a federated learning algorithm with an adaptive server step-size strategy, to resolve the known issue of data heterogeneity across clients. Reviewers anonymously agree that the paper is clearly written. The algorithm is simple and shown to be effective empirically. The experiments in the initial submission had some issues (about the short-runs) as raised by a reviewer, but the revision and rebuttal have addressed the concern. Overall reviewers agree that the experiment are convincing for validating the effectiveness of the FedExp.  For the theoretical analysis of FedExp, the authors first established the connection between FedAvg and the Projection Onto Convex Sets (POCS) algorithm under the overparametrization convex setting. The analysis, although it is based on standard assumptions, is sufficient for the purpose of proving the convergence of FedExp.

**Note From Pc:**

if the above contains the word "oral" or "spotlight" please see: "oral" presentation means -> notable-top-5% and "spotlight" means -> notable-top-25%. As stated in our emails, we are disassociating presentation type from AC recommendations